# Language Models May Verbatim Complete Text They Were Not Explicitly Trained On

**Ken Ziyu Liu** [1 2]   **Christopher A. Choquette-Choo** [* 1]   **Matthew Jagielski** [* 1]   **Peter Kairouz** [1]   **Sanmi Koyejo** [3]
**Percy Liang** [* 3]   **Nicolas Papernot** [* 1]

## Abstract

An important question today is whether a given text was used to train a large language model (LLM). A *completion* test is often employed: check if the LLM completes a sufficiently complex text. This, however, requires a ground-truth definition of membership; most commonly, it is defined as a member based on the $n$-gram overlap between the target text and any text in the dataset. In this work, we demonstrate that this $n$-gram based membership definition can be effectively gamed. We study scenarios where sequences are *non-members* for a given $n$ and we find that completion tests still succeed. We find many natural cases of this phenomenon by retraining LLMs from scratch after removing all training samples that were completed; these cases include exact duplicates, near-duplicates, and even short overlaps. They showcase that it is difficult to find a single viable choice of $n$ for membership definitions. Using these insights, we design adversarial datasets that can cause a given target sequence to be completed without containing it, for any reasonable choice of $n$. Our findings highlight the inadequacy of $n$-gram membership, suggesting membership definitions fail to account for auxiliary information available to the training algorithm.

## 1. Introduction

*Training data membership* asks whether a data point was used to train a given model. For large language models (LLMs), it is useful to answer questions related to privacy (e.g., is the LLM leaking information contained in the text it was trained on?), copyright (e.g., has the model been trained on copyrighted text?), and more generally AI safety (e.g., did the LLM successfully unlearn text that was identified as harmful post hoc training?). In each of these settings, it is important that the evaluation of membership be robust.

With direct access to the training dataset, determining sequence membership is straightforward given a fixed criterion for defining when two text sequences are "the same." A common approach is to compare sequences by their $n$-grams, as this is both efficient and reasonable. Two sequences match *verbatim* if all their $n$-grams are equal for all $n$, and *approximately* if most do for some $n$ (Lee et al., 2021). However, this definition must align with downstream concerns in privacy, copyright, and safety—domains focused on what an LLM can reproduce as an intuitive notion of membership.

Consider a setting where a long text sequence is generated verbatim by an LLM. The sequence has high entropy due to its length, which makes it very unlikely to have been generated by chance. Thus, one may intuitively conclude that this sequence was *in* the training data. This raises a key question: can an LLM generate a target sequence even if it was never included as an $n$-gram in its training data? Our findings confirm that the answer is affirmative: $n$-gram membership establishes a threshold dependent on $n$, and this threshold can be *gamed*. In other words, our findings show that that formally defining a robust and accurate notion of membership is challenging.

We assess whether an LLM generates a target sequence by prompting it with a prefix and checking if it completes the corresponding suffix, a process we refer to as a *completion test*. Language models are known to complete some of their training data (Nasr et al., 2023). In our work, we first find that even after removing a set of extracted sequences from the training dataset and retraining the LLM *from scratch*, the retrained model can still verbatim complete 40% of them under our experimental conditions (Section 4). Upon investigation, we find that these removed yet still completed sequences are either *de facto* members of the training set (but for a different definition of membership) or lacking sufficient complexity: many examples have near duplicates, sequences with $m < n$-grams that are not removed, or are

---

[*]Equal contribution   [1]Google   [2]Work completed while on internship at Google DeepMind.   Now at Stanford University.   [3]Stanford University.   Correspondence to: Ken Liu <kzliu@cs.stanford.edu>, Christopher A. Choquette-Choo <cchoquette@google.com>.

*Proceedings of the $42^{nd}$ International Conference on Machine Learning*, Vancouver, Canada. PMLR 267, 2025. Copyright 2025 by the author(s).

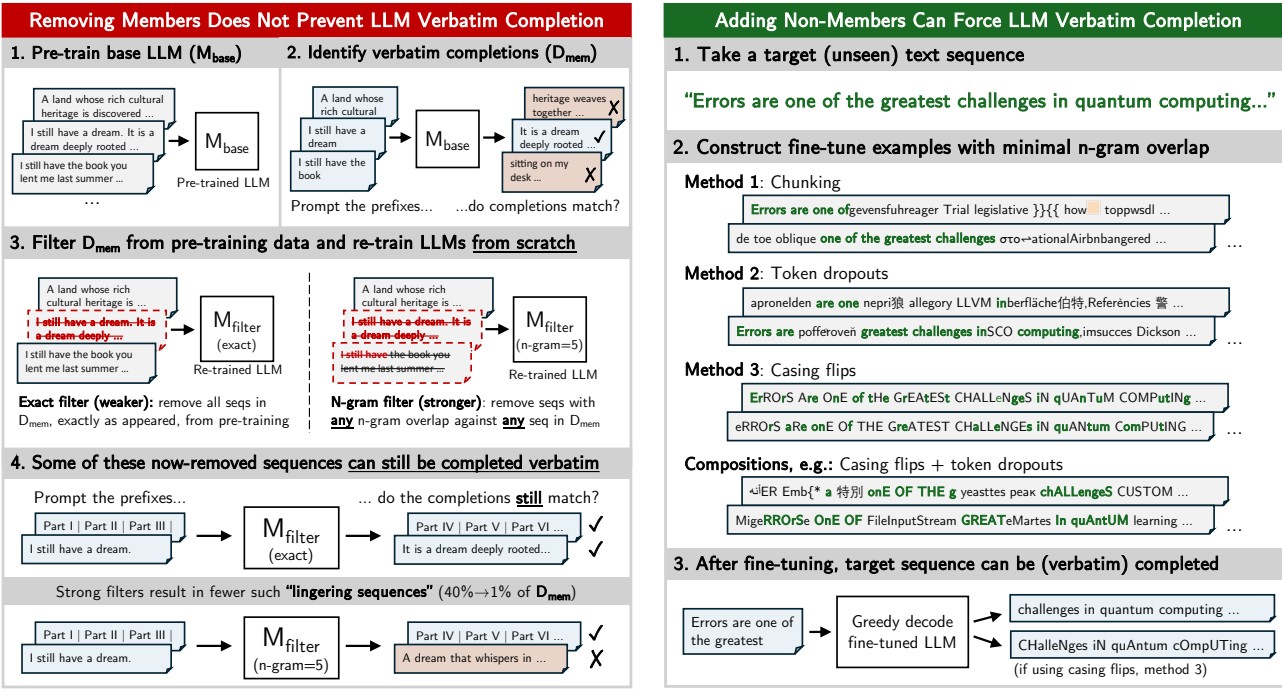

Figure 1: **Main setup and findings**: a text sequence can be (verbatim) completed by a language model without being a $n$-gram "member" of its training set. **Left (§4):** We pretrain a model and remove extracted training dataset of length $k$ from its training data with either exact ($k = n$)-gram filters or stronger approximate ($k > n$)-gram membership filters. We find some sequences ($\approx 40\%$ with exact filters or $\approx 1\%$ with approximate filters) remain verbatim completed despite not being explicitly trained on. **Right (§5):** We show an LLM can be fine-tuned to verbatim complete a target unseen sequence, e.g., today's blog post, by using adversarially constructed datasets with *no* $n$-gram overlap.

explained by the model's generalization capabilities (e.g., patterns or counting).

This result, however, leads to our second finding—there exists strategies for systematically gaming the $n$-gram membership definition. That is, there are strategies for constructing a dataset $D$ that does not contain $n$-grams of a sequence $x$, yet, when a language model trains on $D$ it is able to complete $x$ verbatim. In Section 5, we give multiple examples of such strategies, including one that has the model train on multiple $m$-grams of $x$, where $m < n$ and, in some cases, $m \ll n$. Our experiments show that we can systematically force a model to complete six sequences of interest $x$ despite these sequences not being a member of its training set per the $n$-gram membership definition.

Figure 1 shows our setup. Our main takeaways are:

1. We find that there is high overlap between training data membership and our LLM completion test being positive. Text not in this overlap are explained by the lack of complexity or limitations in $n$-gram based definitions.
2. $n$-gram membership is limited in capturing the intuition of what constitutes a training dataset "member." Indeed, our work shows that a model can complete sequences that are not $n$-gram members of its training dataset.

3. We believe that the underlying cause of this limitation is not in the choice of the distance used to compare sequences—here $n$-gram overlap—but rather in the fact that the membership definition fails to consider auxiliary information that the training algorithm gets access to, e.g., through pre-processing or other design choices made throughout the ML pipeline. Here, the strategies we propose to game the $n$-gram definition exploit this by introducing auxiliary information through the very construction of the training dataset: e.g., we cannot construct $(n-1)$-grams of a sequence $x$ without knowing the entire sequence in the first place.

## 2. Background & Related Work

**Definitions of data membership.** Many language model tasks require a definition of data membership. In most cases, the definition falls into versions of $n$-gram, or substring, overlap (Anil et al., 2023; Gemini Team et al., 2023; 2024; Gemma Team et al., 2024a;b; Touvron et al., 2023; Dubey et al., 2024; Zhang et al., 2024a; Duan et al., 2024; Carlini et al., 2021; Singh et al., 2024). $n$-gram based definitions capture near-duplicates by matching smaller text segments; this is flexible, simple, and intuitive. When studying *data contamination*, much of the prior work uses $n$-gram based

definitions (Sainz et al., 2023; Jiang et al., 2024; Dekoninck et al., 2024; Singh et al., 2024). For example, GPT-4 considers 50-character substring overlap (Achiam et al., 2023) and Llama-3 considers 8-gram token overlap (Dubey et al., 2024). For *training data deduplication* (Lee et al., 2021; Kandpal et al., 2022; Mou, 2023), duplicates are identified based on training data membership. Recent methods use suffix arrays for exact substring matches (Lee et al., 2021) and MinHash or locality sensitive hashing for approximate matches (Broder, 1997; Mou, 2023); both build on $n$-gram overlap. The prevalent use of $n$-gram based definitions reflects a practical balance between accuracy and simplicity. A key focus of our work is to highlight the limitations of these $n$-gram based definitions.

**Tests for data membership.** Unlike membership definitions, which define the ground-truth, membership tests aim to detect if a data sample was in a dataset. There are many model-level membership tests in the literature that predict membership of text to a training dataset with only access to a trained model, and not the training dataset. Our work focuses on model-level membership tests because they are more relevant to the downstream uses of membership in LLMs, e.g., in privacy, copyright, and safety (see §1).

*Membership inference attacks (MIA)* (Shokri et al., 2017) are widely studied, especially in computer vision (Yeom et al., 2018; Salem et al., 2018; Sablayrolles et al., 2019; Choquette-Choo et al., 2021; Carlini et al., 2022a; Jagielski et al., 2024) and more recently example-level membership inference for LLMs (Zarifzadeh et al., 2023; Shi et al., 2023; Mattern et al., 2023; Li et al., 2023). Despite these attempts, progress is hindered by flawed evaluations (Meeus et al., 2024; Zhang et al., 2024b): Duan et al. (2024) argue that membership can be inherently blurry for natural language, Das et al. (2024) report that existing MIA testbeds suffer from distribution shifts, and Kong et al. (2023) refute MIAs using a gradient-space attack. Our work situates in this body of work by studying systematic failure modes of operationalizing membership through definitions and tests, and the consequences when these definitions and tests mismatch.

*Dataset-level* MIAs enhance membership signals by leveraging multiple correlated samples as inputs (Maini et al., 2021; Kandpal et al., 2023; Maini et al., 2024). These are closely related to contamination tests (Golchin & Surdeanu, 2023; Oren et al., 2023). Our work focuses on sequence-level data membership tests based on data completion, because these focus on scenarios where the LLM generates the text, which presents novel concerns for privacy, copyright, and safety.

*Data completion.* There is a long body of work studying generation of training data, in diffusion models (Somepalli et al., 2023; Carlini et al., 2023) and in LLMs (Carlini et al., 2019; Tirumala et al., 2022; Kudugunta et al., 2024; Biderman et al., 2024; Freeman et al., 2024; Huang et al., 2024). These works are often studied from the perspective of studying memorization, where the entity performing the model test has access to the training dataset. In this line of literature, there exist both verbatim definitions of memorization (Carlini et al., 2022b; Huang et al., 2024) and approximate definitions (Ippolito et al., 2022). When studied from a black-box perspective—without access to the training dataset—they typically match completions against known auxilliary databases as a surrogate confirmation of membership (Carlini et al., 2021; Nasr et al., 2023). Intuitively, if a model completes a long sequence $x$ when prompted with its prefix, it likely saw $x$ during training because $x$ has high entropy due to its length and vocabulary size (Carlini et al., 2019; 2022b). Our work focuses only on these completion tests as a black-box membership test.

## 3. Preliminaries

We now formalize the key definitions that underlie our experiments. We focus on defining what it means for a sequence to be a "member" of the training set of a language model, and what constitutes "completing" a sequence as a means of testing its membership. Precise definitions of these notions anchor our study of the mismatch between them.

Modern language models operate on *token sequences*, which are integer encodings of text strings via a byte-pair encoding (BPE) tokenizer (Sennrich, 2015). We use $x$ to denote a token sequence (rather than its text form) with length $|x|$, and n-grams$(x) = \{x_{i:i+n}\}_{i=1}^{|x|-n}$ to denote the set of $n$-grams derived from $x$.

**Definition of *Data Membership*.** We anchor on a simple and flexible membership definition for our experiments that encapsulates many variants used in the literature:

**Definition 3.1** ($n$-gram data membership)**.** *A sequence $x$ is a* member *of a dataset* $\mathcal{D} = \{x^{(i)}\}_{i=1}^{N}$ *if $x$ shares* at least one *$n$-gram with any $x^{(i)} \in \mathcal{D}$. That is, $x$ is* member *if there exists a $g \in$ n-grams$(x)$ s.t. $g \in \bigcup_i$ n-grams$(x^{(i)})$.*

This definition is stringent (e.g., approximate membership typically requires many, not just one, $n$-gram to match). This ensures we overestimate members and thus underestimate non-members. This definition is also inclusive of the those in the literature, as varying $n$ captures a spectrum of them. For example, setting $n = |x|$ is the verbatim membership used in Carlini et al. (2022b). Smaller $n$ captures many approximate membership definitions, such as MinHash (Broder, 1997; Lee et al., 2021), edit distance based membership (Ippolito et al., 2022), and many other $n$-gram variants cited in Section 2. In the remainder of the paper, we call a sequence $x$ an "$n$-gram member" if $x$ satisfies Def. 3.1, and otherwise a "$n$-gram non-member".

**Definition of *Data Completion*.** Informally, we define a *completion* as: when a token sequence is known *a priori*

and a language model generates its suffix when prompted with its prefix. Formally, if $x = [p\|s]$, then model generates the expected suffix $s$ of $x$ based on the provided prefix $p$ (prompt). For simplicity, we focus on $|p| = |s| = |x|/2$ in our experiments. Prior work has studied how the choice in prefix and suffix lengths impact memorization (Carlini et al., 2022b; Huang et al., 2024). To capture highly similar but not verbatim completions of the sequence $x$, we introduce variants of completion that allow for semantically insignificant deviations from the original $s$. We define the following notions of completion.

**Definition 3.2** (Exact completion). *Given tokens $x = [p\|s]$ and a model $\mathcal{M}$, we say $x$ is* exactly completed *if $\mathcal{M}(p) = s$ using greedy decoding.*

This is closely related to verbatim memorization and verbatim training data extraction (see Section 2). There are also semantically equivalent sequences that humans would be likely to not distinguish from the original sequence. We thus consider two approximate notions of completion, relevant to our experiments in Section 5.

**Definition 3.3** (*r*-similar completion). *Given $x = [p\|s]$ and $\mathcal{M}$, we say $x$ is a $r$-similar completion if $\mathcal{M}(p)$ is within a normalized Levenshtein edit distance of $1 - r$ using greedy decoding, i.e., $\mathrm{lev}(\mathcal{M}(p), s)/\max(|\mathcal{M}(p)|, |s|) \leq 1 - r$.*

**Definition 3.4** (Case-insensitive completion). *Given $x = [p\|s]$ and $\mathcal{M}$, we say $x$ is a* case-insensitive completion *if $\mathrm{lower}(\mathcal{M}(p)) = \mathrm{lower}(s)$ with greedy decoding, where $\mathrm{lower}(\cdot)$ applies character-wise lower casing.*

**Data Completion vs. Data Extraction.** Data *extraction*, as considered in recent work (Carlini et al., 2021; Nasr et al., 2023), concerns recovering *training* data from the model. This thus involves both (1) data *completion* (e.g., as in Def. 3.2), and (2) verifying that the completion is a training *member*; e.g., by inspecting the training data. In a sense, extraction specifically measures *memorization*,[1] while completion is more generic—indeed, our work studies *non-member* completions; Fig. 2 illustrates the distinction.

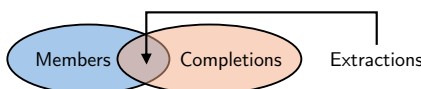

Figure 2: We say that a sequence is *extractable* if it can be *completed* and can be proved a *member* of the training set.

---

[1]Our reference to "memorization" is intended in a very specific context: whether a model can be induced to generate near-copies of some training examples when prompted with appropriate instructions, and often with prior knowledge of the model's training data. Specifically, we do not mean to imply that a model "contains" its training data in the sense that any arbitrary instance of that data can be retrieved without use of specialized software or algorithms. Rather, if a model can be induced to generate very close copies of certain training examples by supplying appropriate instructions to guide the model's statistical generation process then that model is said to have "memorized" those examples.

## 4. Removing Members Does Not Always Prevent LLM Verbatim Completion

LLMs are known to complete training sequences (Carlini et al., 2021). In this section, we ask:

> **Main Question**: Will an LLM still complete a text sequence even after we *remove* all training sequences that have $n$-gram overlap with it?

We find that this retrained model still has *lingering completions*: the completion test still succeeds despite having removed all completions identified by $n$-gram matching from the training data. This raises the question: why is the LLM still completing these sequences? As we will see, this is because the completions are either a) still contained in the dataset but via a different membership definition (e.g., for some $m$-gram membership, with $m < n$), or b) they lack sufficient entropy and can be easily predicted (generalized).

### 4.1. Experimental Setup

Our high-level experimental framework is as follows. We defer additional details to Appendix A.

1. **Pre-train a base model**: We first pre-train a standard LLM $\mathcal{M}_{\text{base}}$ from scratch on a training dataset $\mathcal{D}_{\text{base}}$.
2. **Identify verbatim completions**: We then collect a set of sequences $\mathcal{D}_{\text{mem}}$ of length $k$ that $\mathcal{M}_{\text{base}}$ can complete *verbatim* (as in Def. 3.2), by checking the first $k$ tokens of *every* training document in $\mathcal{D}_{\text{base}}$. This is a simple and effective procedure since LLMs are known to memorize training data (e.g., Carlini et al. (2022b)); other choices to obtain $\mathcal{D}_{\text{mem}}$ are also possible.
3. **$n$-gram filtering**: We then filter each sequence $x \in \mathcal{D}_{\text{mem}}$ away from $\mathcal{D}_{\text{base}}$. Our filtering procedure is simple and reflects $n$-gram membership (Def. 3.1): to filter a sequence $x$, we perform a sliding window of length $|x|$ over $\mathcal{D}_{\text{base}}$ (as if all tokens are concatenated into a single array); if the window shares any $n$-gram with n-grams($x$), the window is deleted from the pre-training data. When $n = |x|$, then we filter $x$ exactly as it appears in the dataset; when $n < |x|$, we filter more aggressively as the window is removed on partial matches against $x$. The filtered dataset is denoted as $\mathcal{D}_{\text{filter}}^{(n)}$.
4. **Re-train a counterfactual model**: Pre-train another LLM $\mathcal{M}_{\text{filter}}^{(n)}$ *from scratch* on the filtered data $\mathcal{D}_{\text{filter}}^{(n)}$.

We repeat this procedure for different model sizes, different values of $n$-gram (for filtering), and different sequence lengths $k$. Unless otherwise stated, we use $k = 50$, meaning that for a sequence $x = [p\|s]$, we have $|p| = |s| = 25$. We provide results on $k = 100$ in Appendix A.7.

**Models and training.** We pre-train a series of models from scratch using the GPT-2 architecture (Radford et al., 2019), spanning sizes of {350M, 774M, 1.6B, 2.8B} parameters,

Table 1: The number of identified verbatim memorized sequences $|\mathcal{D}_{\text{mem}}|$ at different model sizes (step #2 of § 4.1).

| Model size | 304M | 774M | 1.6B | 2.8B |
|---|---|---|---|---|
| $|\mathcal{D}_{\text{mem}}|$ | 76,648 | 116,270 | 151,598 | 175,813 |

Figure 3: **LLMs can verbatim complete texts with zero $n$-gram overlap to training data.** A fraction of sequences filtered away from pre-training data ($\mathcal{D}_{\text{mem}}$, Table 1) can still be completed by the re-trained LLM verbatim. The fractions decrease under stronger filtering (smaller $n$-gram filter) and remain relatively stable across model scales.

with 1.6B being the size of the original GPT-2 XL and 2.8B being a scaled-up model. We use LLM.c (Karpathy, 2024) for an efficient pre-training pipeline. We primarily report results on the 1.6B model unless otherwise stated.

**Data.** For all models, we use FineWeb-Edu (Penedo et al., 2024) as a state-of-the-art pre-training dataset.[2] We use the same base dataset $\mathcal{D}_{\text{base}}$ of 33.6B randomly sampled tokens. For the 1.6B model, 33.6B tokens is approximately Chinchilla optimal ($\approx 20$ tokens per parameter, Hoffmann et al. (2022)). For consistency, we train the base models $\mathcal{M}_{\text{base}}$ of different sizes with the same starting dataset $\mathcal{D}_{\text{base}}$; as the size of $\mathcal{D}_{\text{mem}}$ (step #2) hinges on the size of $\mathcal{M}_{\text{base}}$ (Carlini et al., 2022b), we obtain different filtered datasets $\mathcal{D}_{\text{filter}}^{(n)}$ for each model size (by inferencing on $\mathcal{M}_{\text{base}}$).

### 4.2. Results

With the artifacts $\mathcal{M}_{\text{base}}, \mathcal{D}_{\text{base}}, \mathcal{D}_{\text{mem}}, \mathcal{D}_{\text{filter}}^{(n)}, \mathcal{M}_{\text{filter}}^{(n)}$, we now make observations pertaining to our main question.

**Finding #1 (Existence of Lingering Sequences): LLMs can *verbatim* complete a fraction of the sequences deleted from training data, and consistently so across scale.** On a macroscopic level, we first observe that simply deleting a set of sequences from pre-training data does *not* always prevent them from being generated by an LLM (Fig. 3). This observation is consistent across model scales, where each size has a different amount of memorization (Table 1). We call these "*lingering sequences*" and denote them as $\mathcal{D}_{\text{linger}}^{(n)}$. Under our experimental conditions, the frac-

tion of lingering sequences $|\mathcal{D}_{\text{linger}}^{(n)}|/|\mathcal{D}_{\text{mem}}|$ can be as high as 40% when we apply the weakest $n$-gram filter and only remove verbatim sequence matches ($n = 50$).

**Finding #2 (Nature of Lingering Sequences): We found no lingering sequences that correspond to *creative* generalization—sequences beyond reconstructions from neighboring texts and continuations of recognizable patterns.** Fig. 4 visualizes a few lingering sequences in $\mathcal{D}_{\text{linger}}^{(n)}$ and see Appendix A.2 for more. To understand their origin, we then perform a search of neighboring texts (Levenshtein distance $< 20$) for a few randomly[3] selected lingering sequences over the large pre-training data $\mathcal{D}_{\text{base}}$; we defer results to Appendix A.3. For all lingering sequences we queried, we were able to find near-duplicates, yet all such copies evaded $n$-gram overlap detection one way or another. This result sheds light on the remarkable ability for LLMs to generalize from neighboring text. More interestingly, it also informs an interesting symmetry on how we may *adversarially* construct training sequences that: (1) have no $n$-gram overlap with a target sequence $x$, and (2) yet serve as "anchor points" that the LLM can interpolate to verbatim complete $x$. We explore this symmetry in the coming section (§5).

**Finding #3 (Persistence of Lingering Sequences): Stronger filters reduce, but do not eliminate, these lingering sequences, and instead shift their distribution to more generalizable patterns.** As we filter $\mathcal{D}_{\text{mem}}$ from $\mathcal{D}_{\text{base}}$ more aggressively with smaller $n$-gram filters, smaller fractions of $\mathcal{D}_{\text{mem}}$ can be completed by the re-trained model $\mathcal{M}_{\text{filter}}^{(n)}$ verbatim (Fig. 3). However, even at a very conservative filter of $n = 5$ (a sequence is removed from $\mathcal{D}_{\text{base}}$ if any 5-gram is in $\mathcal{D}_{\text{mem}}$), $\mathcal{D}_{\text{linger}}^{(5)}$ still accounts for $\approx 1\%$ of $\mathcal{D}_{\text{mem}}$ (Table 1). As the fraction of lingering sequences decreases, their contents also shift from verbatim memorization of semantically useful text (e.g., famous quotes) to generalizable patterns (e.g., counting in Roman numerals). We provide examples in Fig. 4 and Appendix A.2.

To quantify this shift, we use three proxy metrics (Fig. 5), though we note that none perfectly captures the (fuzzy) boundary between memorization and generalization. First, we measure the verbatim completion rate of $\mathcal{D}_{\text{linger}}^{(n)}$ using the off-the-shelf GPT-2-XL (Radford et al., 2019); since $\mathcal{M}_{\text{filter}}^{(n)}$ is a similar model by construction, a lingering sequence is likely a generalizable pattern if both models (trained on distinct data) agree on its completion. Second, we similarly consider the completion rate of a counterfactual model $\mathcal{M}_{\text{cf}}$ on pre-training shards disjoint from $\mathcal{D}_{\text{base}}$. Third, we prompt Gemini 1.5 Pro with few-shot examples to determine if a lingering sequence is a pattern continuation (prompt template in Appendix A.10). All proxy metrics confirm

---

[2]This work may contain information from FineWeb-Edu dataset, which is made available under the ODC Attribution License.

[3]We only perform this experiment on randomly selected lingering sequences due to the cost of the search.

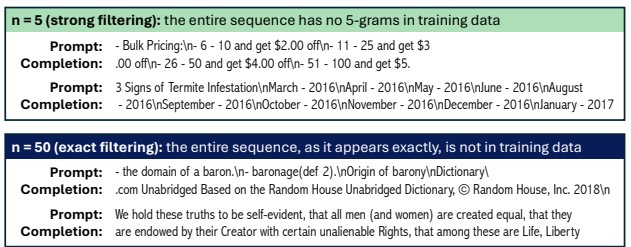

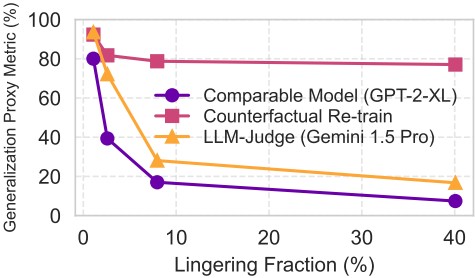

Figure 4: Examples of lingering sequences (more in A.2).

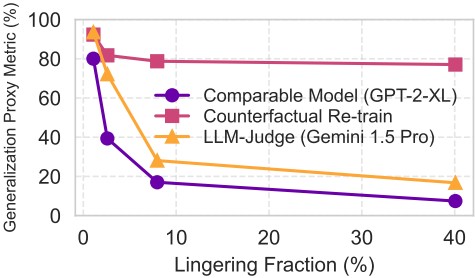

Figure 5: **Strong filters result in less lingering completions and shift them toward generalizable patterns.** Results on 1.6B size. We use three proxy metrics where higher indicates more pattern continuations: 1) % verbatim completion by off-the-shelf GPT-2-XL; 2) % verbatim completion by $\mathcal{M}_{cf}$, a counterfactual model trained on disjoint pretraining shards; 3) % judged as patterns by Gemini 1.5 Pro.

our manual inspection that stronger filters force out generalization behaviors from the model, albeit these are simple sequences to generalize to (recall finding #2).

### 4.3. Interpretations and Outlook

While we found no evidence of creative generalization in models up to 2.8B parameters, lingering sequences are intriguing because they seem to challenge our understanding of membership in LLMs—if a language model can verbatim complete sequences known *a priori* without ever training on any of its $n$-grams, what does this imply for the definition of membership and its reliance on $n$-gram overlap? To better understand these limitations we identified—and if they are exclusively explained by the limitation of $n$-gram overlaps—we next experiment with adversarially gaming $n$-gram membership. We build on our findings thus far to inform the adversarial construction of a dataset of $n$-gram non-members that is able to force LLM completion.

## 5. Adding Non-Members Can Force LLM Verbatim Completion

While lingering sequences (Section 4) are largely benign and rare, observing how they came to be (e.g., through visualizations in Appendix A.3) informs how one may *adversarially* force a model to complete $n$-gram non-members. We explore this direction with the following:

**Main Question**: Given a chosen (unseen) text sequence $x$, can we add training sequences $\mathcal{D}_{ft}$ that have no $n$-gram overlap with $x$, and yet an LLM fine-tuned on $\mathcal{D}_{ft}$ can complete $x$ verbatim?

This question is intriguing for its insights into LLM generalization, as well as its relevance to *adversarial* manipulation of training data, where an adversary may wish to intentionally avoid $n$-gram membership for, e.g., harder-to-detect data poisons and content misuse (more in Section 5.6).

To answer the question, consider a class of *noisy transformations* $f$ on $x$, such that: (1) $\tilde{x} = f(x)$ retains some information about $x$; and (2) $\tilde{x}$ has no $n$-gram overlap with $x$. We hypothesize that an LLM trained on different instances of $\tilde{x}$ (over different randomness) should learn to recover the original $x$, much like a denoising antoencoder learning to recover clean data from noisy inputs.

**Methods.** We show that it is possible to adversarially construct examples $\mathcal{D}_{ft} = \{\tilde{x}_i\}$ from a chosen *unseen* example $x$ (e.g., today's news) such that $x$ and $\mathcal{D}_{ft}$ share no common $n$-grams (and many membership tests, including manual inspection, would fail). Yet with only a few gradient steps of fine-tuning an LLM on $\mathcal{D}_{ft}$, the model can (verbatim) complete this "non-member" $x$. We study three such methods of constructing such a fine-tuning set $\mathcal{D}_{ft}$:

1. *Stitching chunks*: We split $x$ into overlapping segments padded with random tokens.
2. *Token dropouts*: We replace tokens in $x$ at different (random) positions with random tokens; positions have at most $n - 1$ gap to avoid $n$-gram overlaps.
3. *Casing flips*: We flip the casing of every English letter in $x$ with probability $p$.

These methods resemble real-world transformations of texts such as taking excerpts of an article and transcriptions that misspell words, miss punctuation, and drop casing. They also have varying degrees of efficacy as we will discuss in Section 5.5. Note that our goal is not to find the best possible (stealthiest) transformation, but to explore feasibility and ease of such adversarial manipulation.

**Models and training.** We work with two model families: Gemma-2 (Gemma Team et al., 2024a;b) and Qwen-2.5 (Yang et al., 2024; Team, 2024), spanning model size from 0.5B to 9B. We fine-tuned these models to predict the next token with a batch size of 32 and a constant learning rate of $10^{-5}$.

**Data.** We primarily experiment on three target texts. These texts are all roughly 1,000 characters long ($\approx 250$ tokens under Gemma-2 tokenizer) and have a recent temporal cutoff such that they could not have been included in the training set of Gemma-2 and are extremely unlikely to appear in the training set of Qwen-2.5 (thus helps ablate the effect of

potential memorization):

1. **Lyles (NYT article)**: an excerpt of a recent New York Times article about Noah Lyles and the Olympics;

2. **Karpathy (tweet)**: a tweet text in an image posted by Andrej Karpathy about LLM tokenization; and

3. **Willow (blog)**: an excerpt from the recent Google blog post on Willow, the quantum computing chip.

The text choices are otherwise arbitrary and alternatives are possible; see the full texts, source, and results on alternative texts in Appendix B.1. For every target text sequence, we construct $N = 2,000$ examples as $\mathcal{D}_{ft}$ by applying the transformation $f$ with different randomness, though in most settings we need less than 1,000 examples. We visualize some of these examples in Appendix B.3.

### 5.1. Stitching chunks

A natural way to avoid generating any $\tilde{x}$ that shares $n$-grams of $x$ is to only expose chunks of at most $(n-1)$-grams to the model. We construct $\mathcal{D}_{ft}$ by breaking $x$ into contiguous, overlapping token segments, with the remaining positions padded with random tokens from the vocabulary. This procedure is parameterized by the chunk size $c$ and the overlap $l$. Chunk size $c$ controls the difficulty (noise level) of the task. For example, the task is trivial if $c$ is the sequence length (since $\mathcal{D}_{ft}$ are just copies of $x$). A small $c$ means most tokens of any $\tilde{x} \in \mathcal{D}_{ft}$ are random. A large overlap $l$ should intuitively help the LLM learn to stitch $x$ together, though empirically we observe minimal impact (Appendix B.5.1).

To illustrate, if the target $x = [1, 2, 3, 4, 5, 6]$, then we may have $\mathcal{D}_{ft} = \{[1, 2, 3, \cdot, \cdot, \cdot], [\cdot, \cdot, 3, 4, 5, \cdot], [\cdot, \cdot, \cdot, \cdot, 5, 6], ...\}$ with chunk size $c = 3$ and overlap $l = 1$, ($\cdot$ denotes a fresh random token). Intuitively, the task for the LLM is to "stitch" the token chunks back together into $x$. A similar technique is explored in the concurrent work of Panaitescu-Liess et al. (2024) for data poisoning; here, we explore chunk sizes, overlaps, and model families to present more comprehensive tradeoffs. See Algorithm 1 for our detailed procedure and Appendix B.3 to visualize examples in $\mathcal{D}_{ft}$.

### 5.2. Token dropouts

Another way to avoid $\tilde{x} \in \mathcal{D}_{ft}$ sharing $n$-grams with $x$ is to mask out tokens in $\tilde{x}$ at least every $n$ positions, so that it cannot share $n$-gram overlap with $x$. That is, $\mathcal{D}_{ft}$ contains different versions of $x$ where at least every $(n-1)$-th token is masked out (replaced with a random token) so that there is at most $n - 1$ token overlap. This procedure is parameterized by a drop interval $d$, representing the length of the interval between dropped tokens. It must be that $d \leq n$ to ensure no $n$-gram overlap. To illustrate, if $x = [1, 2, 3, 4, 5, 6]$, then we may have $\mathcal{D}_{ft} = \{[1, 2, 3, \cdot, 5, 6], [1, \cdot, 3, 4, 5, \cdot], [\cdot, 2, 3, 4, \cdot, 6], ...\}$, where '$\cdot$'

is a fresh random token and where here $d = 4$. We also consider a *randomized* dropout, where every token is dropped with probability $1/d$. By construction, the deterministic version guarantees that $x$ is not a $d$-gram member of the training set, while the randomized version does so with (exponentially) high probability. Notably, due to BPE tokenization (Sennrich, 2015), the original text becomes visually obfuscated for humans even when most tokens are retained ($d > 2$). See Algorithm 2 for detailed algorithm and Appendix B.3 for visualization.

This construction is closely related to the *goldfish loss* proposed by Hans et al. (2024) to mitigate verbatim memorization, where the loss of every $n$-th token (on average, if randomized) is omitted during training. However, the procedure here is entirely *data-centric*: it does not interfere with the training objective and makes the learning task harder as subsequent tokens would still attend to the random tokens.

### 5.3. Case flipping

Another approach is to perform text-space transformations that preserve semantics yet drastically alter the token-space representations. One such method is to randomly flip the *casing* of English letters, creating varied *tokenization* of the otherwise equivalent string. To illustrate, if $x$ decodes to 'This is a string', $\mathcal{D}_{ft}$ may include token sequences of strings like 'THIS Is A stRinG'. Due to the mechanisms of BPE tokenization, it is extremely easy to obtain $\tilde{x}$ with *completely distinct* tokens than $x$ under modern LLM tokenizers (see Appendix B.2 for visualization). The case flipping procedure is parameterized by the flip probability $p$. $p = 0.5$ creates the highest variance, and $p$ closer to 1 always flips the case (most letters being initially lower case, natural text becomes mostly upper case).

### 5.4. Compositions

The transformations presented earlier are not mutually exclusive, and in principle they can be composed arbitrarily. We explore one such composition of token dropouts & case flipping, where non-dropped tokens (§5.2) have casing randomly flipped in the text space (§5.3). A key benefit of compositions is that they combinatorially give rise to many new transformations with potentially more granular control of task difficulty and detectability (e.g., whether manually inspecting $\mathcal{D}_{ft}$ reveals $x$; see Appendix B.3 for example visualizations). We leave a comprehensive evaluation of composed transformations to future work.

### 5.5. Results

**Finding #1: It is possible for an LLM to complete an *unseen* string with no $n$-gram membership after minimal finetuning.**

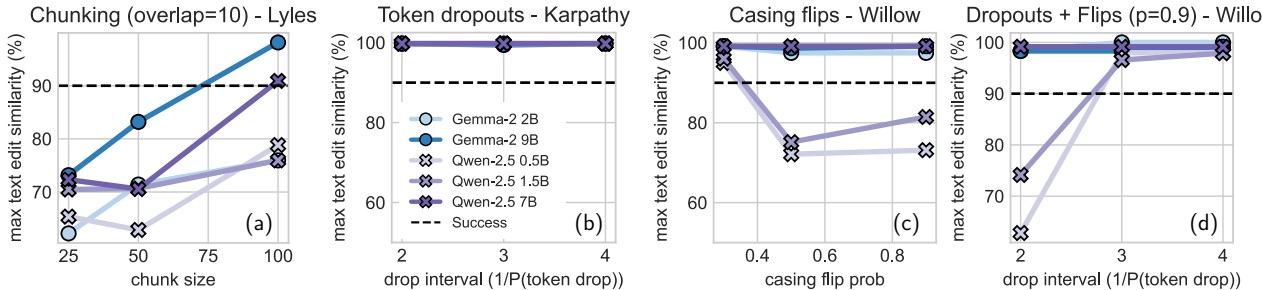

Figure 6: **Completion success across methods and target texts:** (a) chunking on **Lyles**, over chunk size $c$ ($x$-axis); (b) token dropouts on **Karpathy**, over drop interval $d$; (c) casing flips on **Willow**, over flip prob $p$ ($p = 0.5$ is noisiest); and (d) combining dropouts + flips on **Willow**. We observe that: (1) it is possible to complete a chosen string with zero $n$-gram membership, and (2) this ability improves with model size. See Appendix B.5 for comprehensive results.

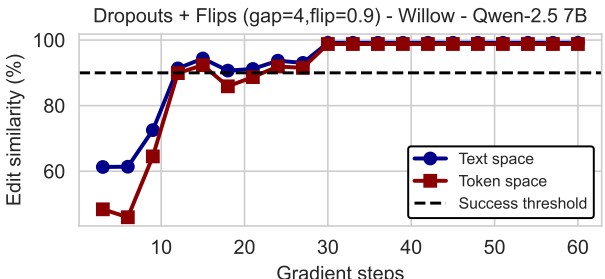

Figure 7: **Completion success may only require a few gradient steps.** See more configurations in Appendix B.5.

Fig. 6 shows the results of various transformations across various text targets (see Appendix B.5 for comprehensive results). We first see that there are many configurations where fine-tuning on $\mathcal{D}_{\text{ft}}$ allow the successful completion of target $x$ (edit similarity > 90%).

Some transformations are more effective than others. We found that *chunking* (Section 5.1) was ineffective: with a small chunk size ($c = 25$), the models mostly fail to complete the target, and only some models succeed at $c = 100$ (Fig. 6 (a)). *Token dropouts* (Section 5.2), on the other hand, is extremely effective—even the smallest model (Qwen-2.5 0.5B) easily completes the target verbatim at a drop interval of 2 (50% drop probability at every token; Fig. 6 (b)). Our results thus present a counter-case to *goldfish loss* (Hans et al., 2024), as models can still complete targets verbatim when there are multiple versions of the target with *different* token dropout positions (e.g., due to near-duplicates, related to the findings of Section 4). *Casing flips* (Section 5.3) are also generally effective (Fig. 6 (c)). Composing token dropouts and casing flips (Section 5.4) increases task difficulty (fewer successes with small models) but otherwise similarly enables verbatim completion.

In essence, these experiments demonstrate that $n$-gram based membership definitions can be vulnerable to *adversarial* manipulation: the fine-tuning set $\mathcal{D}_{\text{ft}}$ clearly contains information about the text $x$, but when given $x$ and a choice of $n$, it is easy for an adversary to bypass detection and yet have the model generate $x$ verbatim.

**Finding #2: Completion success scales with model size.** Another message from Fig. 6 is that as we increase in the model size, the completion success generally improves under the *same* configurations. This provides evidence that frontier models should be more capable at synthesizing $n$-gram non-members into the target texts.

### 5.6. Interpretations and Outlook

We briefly describe the potential implications of our adversarially constructed fine-tuning datasets:

- **Data poisoning**: $n$-gram non-members of a poison text $x$ can be added to the training set and still induce the generation of $x$. The concurrent work of Panaitescu-Liess et al. (2024) explores adding copyrighted materials as poisoned data with a similar chunking technique (§5.1).
- **Data contamination**: a dishonest model developer may game model evaluations through deliberate data contamination while evading $n$-gram based detection.
- **Reporting train-test overlap metrics**: More broadly, a model developer may self-report train-test overlap statistics (e.g., as part of contamination analysis seen in Dubey et al. (2024); Gemini Team et al. (2023); Brown (2020)). Our results highlight that it is desirable that developers report additional metrics beyond $n$-gram overlap.

## 6. Concluding Remarks

Lingering sequences (§ 4) and adversarially constructed fine-tuning datasets (§ 5) demonstrate the remarkable ability of LLMs to generalize from neighboring text. They are thus a valuable tool for evaluating LLM capabilities as models and pre-training datasets scale up. We conclude our work with a discussion of the implications of our findings:

**Membership definitions and tests should incorporate new similarity measures**. We showed $n$-gram based membership emits false negatives that may not capture human intuition nor the pragmatic concerns of the copyright, privacy, and AI safety community. On the flip side, tests like mem-

bership inference should consider broader notions of membership beyond individual sequences: a unit of data could be a collection of sequences grouped by similarity (Kandpal et al., 2023; Maini et al., 2024; Cooper et al., 2024).

**Machine unlearning alone is insufficient to address data permissibility concerns in output suppression.** It is widely accepted that a golden baseline of machine unlearning is to retrain a model from scratch without the target forget data (Bourtoule et al., 2021; Liu, 2024; Liu et al., 2024; Cooper et al., 2024). Yet, our experiments perform precisely this counterfactual and reveal that some excluded sequences can still be verbatim generated (Fig. 1, §4). We thus caution that unlearning alone may not always prevent a model from generating a sequence of interest (e.g., a harmful sequence). This is also known as output suppression and is a common goal of unlearning (Cooper et al., 2024).

**Exploring the connection between our work and forging may help more precisely characterize threat models for when completion can serve as evidence of membership.** Readers familiar with the forging (Thudi et al., 2022) literature will have noticed a connection with our work. Forging a step of gradient descent computed on a given minibatch is done by (adversarially) constructing a different minibatch that will result in the same gradient being computed. Perhaps surprisingly at first, gradients can be forged using *non-overlapping* datasets sampled from "natural" distributions. Rather than forging gradients, our methods can be viewed as attempting to forge model outputs. While our work begins to show that there may exist threat models where completion is insufficient evidence for $n$-gram membership, our experiments did not succeed in obtaining a forge in model outputs using "natural" data (recall § 4). We believe this is a valuable direction for future work.

## Impact Statement

Our work studies training data membership in the context of language models. The impact of our work is described in Sections 4.3, 5.6, and 6. As we described earlier, the limitations we identified in $n$-gram based definitions of membership have implications for copyright, privacy, and AI safety. Broadly speaking, our work advocates for additional membership definitions to help better capture human intuition and make membership more pragmatic.

## Acknowledgements

KZL would like to thank Xinran Zhao, Nikil Selvam, Steven Cao, Jing Huang, Yangjun Ruan, Aryaman Arora, Harshit Joshi, Rylan Schaeffer, Chenglei Si, Yanzhe Zhang, as well as members of the p-lambda lab and STAIR lab at Stanford University for helpful discussions and feedback. We thank Milad Nasr and Nicholas Carlini for helpful discussions, comments, and critiques throughout the work.

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

# A. Removing Members Does Not Always Prevent LLM Verbatim Completion (§4)

## A.1. Data filtering

The $n$-gram filter discussed in Section 4.1 can be defined as follows:

**Definition A.1** ($n$-gram data filtering). *Let $\mathcal{F}$ be a set of $n$-grams to filter against. Let $f_n(x, \mathcal{F})$ be the condition that returns 1 if any $n$-gram of the sequence $x$ is in $\mathcal{F}$ (0 otherwise). An $n$-gram filter against $\mathcal{F}$ on a set of sequences $\mathcal{D}$ is thus constructing $\tilde{\mathcal{D}}_n = \{x \in \mathcal{D} \mid f_n(x, \mathcal{F}) = 0\}$.*

Intuitively, a stronger $n$-gram filter (smaller $n$) means that we are removing a sequence on increasingly smaller partial matches against the filter set. In practice, since we are filtering many sequences ($\mathcal{D}_{\text{mem}}$) at once, it is more efficient to build a shared $n$-gram filter hash set from all sequences in $\mathcal{D}_{\text{mem}}$ and then apply a sliding window filtering procedure.

Table 2 shows the amount of tokens kept after applying $n$-gram filtering at different filtering strengths (for sequence length 50, the default setting used in experiments unless otherwise stated). Table 3 similarly shows the amount of tokens for sequence length 100.

Table 2: Fraction of tokens kept after applying $n$-gram filter to sequence length $k = 50$.

| $n$-gram filtering strength | $n = 5$ | $n = 10$ | $n = 20$ | $n = 50$ (Exact Filtering) |
|---|---|---|---|---|
| **Approx % of tokens kept** | 0.6905 | 0.9838 | 0.9938 | 0.9995 |

Table 3: Fraction of tokens kept after applying $n$-gram filter to sequence length $k = 100$.

| $n$-gram filtering strength | $n = 10$ | $n = 20$ | $n = 50$ | $n = 100$ (Exact Filtering) |
|---|---|---|---|---|
| **Approx % of tokens kept** | 0.9820 | 0.9930 | 0.9970 | 0.9995 |

## A.2. Visualizing Lingering Sequences $\mathcal{D}_{\text{linger}}^{(n)}$

In this and the following section, we provide visualizations to the key results described in Section 4 to help develop intuition on our findings.

Recall from §4 that $\mathcal{D}_{\text{linger}}^{(n)}$ refers to the set of lingering sequences that are still verbatim completable by the re-trained from scratch LLM after applying $n$-gram filter (Def. A.1) on the identified memorized sequences $\mathcal{D}_{\text{mem}}$.

Table 4, Table 5, Table 6, and Table 7 show 15 randomly sampled lingering sequence at filtering strengths $n = \{50, 20, 10, 5\}$, respectively. Observe that, as noted in Section 4.2, with stronger filtering strength (smaller $n$), the content gradually shift from semantically useful content to pattern continuations.

Table 4: **Randomly sampled lingering sequences at filtering strength $n = 50$ (exact) filter.** Sequence length $k = 50$.

| Idx | Lingering sequences at filtering strength $n = 50$ (exact) filter. |
|---|---|
| 0 | **Prompt:** 'Common Name: bleeding heart Type: Herbaceous perennial Native Range: Eastern United States Zone: 3 to 9 ' |
| | **Completion:** 'Height: 1.00 to 1.50 feet Spread: 1.00 to 1.50 feet Bloom Time:' |
| 1 | **Prompt:** '—1477 by topic— —Arts and science— —Birth and death categories— —Births –' |
| | **Completion:** ' Deaths— —Establishments and disestablishments categories— —Establishments – Disestablishments— —' |
| 2 | **Prompt:** 'Charcot Joint (Neuropathic Arthropathy) Medicine Central™ is a quick-consult mobile and' |
| | **Completion:** ' web resource that includes diagnosis, treatment, medications, and follow-up information on over 700 diseases and disorders, providing fast answers' |
| 3 | **Prompt:** 'Mienert-barth Surname History The family history of the Mienert-barth last name is' |
| | **Completion:** ' maintained by the AncientFaces community. Join the community by adding to to our knowldge of the Mienert-' |
| 4 | **Prompt:** 'Instructional Supports and Resources Dyslexia is a specific learning disability that is neurological in origin. It is characterized' |
| | **Completion:** ' by difficulties with accurate and/or fluent word recognition and by poor spelling and decoding abilities. These difficulties typically result from a deficit' |
| 5 | **Prompt:** 'Publisher description for Writers at work. The short composition / Ann O. Strauch. Bibliographic record and links to' |
| | **Completion:** ' related information available from the Library of Congress catalog Information from electronic data provided by the publisher. May be incomplete or contain other' |
| 6 | **Prompt:** 'Create healthcare diagrams like this example called Anencephaly in minutes with SmartDraw. SmartDraw includes 1000s of professional healthcare' |
| | **Completion:** ' and anatomy chart templates that you can modify and make your own. Text in this Example: Anencephaly is' |

| Idx | Lingering sequences at filtering strength $n = 50$ (exact) filter. | |
| --- | --- | --- |
| 7 | **Prompt:** | 'Presentation on theme: "Spiraled Assignments Presenter: Angela Pritchett November 14, 2006."' |
| | **Completion:** | '— Presentation transcript: Spiraled Assignments Presenter: Angela Pritchett November 14, 2006 ' |
| 8 | **Prompt:** | 'An excerpt from www.HouseOfNames.com archives copyright © 2000 - 2013 Where did the Irish McSweeney family' |
| | **Completion:** | ' come from? What is the Irish McSweeney family crest and coat of arms? When did the McSweeney family first' |
| 9 | **Prompt:** | 'Which of the following cubes can be made from these nets? Is it possible to remove ten unit cubes from a 3 by' |
| | **Completion:** | ' 3 by 3 cube made from 27 unit cubes so that the surface area of the remaining solid is the same as the surface area' |
| 10 | **Prompt:** | 'Course Hero. "The Libation Bearers Study Guide." Course Hero. 23 June 2017. Web. 14 Nov. 2018' |
| | **Completion:** | '. ¡https://www.coursehero.com/lit/The-Libation-Bearers/¿. Course Hero' |
| 11 | **Prompt:** | 'Presentation on theme: "Cause and Effect Comprehension Skill Fourth Grade Unit 2 Week 1 Created by Kristi Waltke' |
| | **Completion:** | '."— Presentation transcript: Cause and Effect Comprehension Skill Fourth Grade Unit 2 Week 1 Created by Kristi Walt' |
| 12 | **Prompt:** | 'An excerpt from www.HouseOfNames.com archives copyright © 2000 - 2015 Where did the English Ragsdale family' |
| | **Completion:** | ' come from? What is the English Ragsdale family crest and coat of arms? When did the Ragsdale family first' |
| 13 | **Prompt:** | 'Gibbous Scorpio Moon phase on 29 March 2051 Wednesday is Waning Gibbous, 16' |
| | **Completion:** | ' days old Moon is in Scorpio.Share this page: twitter facebook linkedin Previous main lunar phase is the Full Moon' |
| 14 | **Prompt:** | 'What does AIDS mean in Laboratory? This page is about the meanings of the acronym/abbreviation/shorthand' |
| | **Completion:** | ' AIDS in the Medical field in general and in the Laboratory terminology in particular. Find a translation for AIDS in other languages:' |
| 15 | **Prompt:** | 'Nathalie Raphaëlle June 23, 2021 Worksheets If you home school your children, you' |
| | **Completion:** | ' will quickly realize how important printable homeschool worksheets can be. If you are trying to develop a curriculum for your' |
| 16 | **Prompt:** | 'Course Hero. "The Pearl Study Guide." Course Hero. 14 Dec. 2017. Web. 24 Nov. 2020. ¡' |
| | **Completion:** | 'https://www.coursehero.com/lit/The-Pearl/¿. Course Hero. (2017, December' |
| 17 | **Prompt:** | 'Internet of Things Internet of Things The Internet of Things (IoT) is a system of interrelated computing devices' |
| | **Completion:** | ', mechanical and digital machines, objects, animals or people that are provided with unique identifiers and the ability to transfer data over a' |
| 18 | **Prompt:** | 'Latest Newland photos These photos were uploaded by members of the Newland community on AncientFaces. Newland S' |
| | **Completion:** | 'urname History The family history of the Newland last name is maintained by the AncientFaces community. Join the community' |
| 19 | **Prompt:** | 'Definition of Seckles 1. seckle [n] - See also: seckle Click the following' |
| | **Completion:** | ' link to bring up a new window with an automated collection of images related to the term: Seckles Images Lexic' |

Table 5: **Randomly sampled lingering sequences at filtering strength $n = 20$ filter.** Sequence length $k = 50$.

| Idx | Lingering sequences at filtering strength $n = 20$ filter. | |
| --- | --- | --- |
| 0 | **Prompt:** | 'Presentation on theme: "MAKING BOOKS WITH CHILDREN Picture It! Publish It! Read It!"' |
| | **Completion:** | '— Presentation transcript: MAKING BOOKS WITH CHILDREN Picture It! Publish It! Read It!' |
| 1 | **Prompt:** | 'This Constitution, and the Laws of the United States which shall be made in Pursuance thereof; and all Treaties made' |
| | **Completion:** | ', or which shall be made, under the Authority of the United States, shall be the supreme Law of the Land; and' |
| 2 | **Prompt:** | 'How To Recognize A Crystal Child A selection of articles related to how to recognize a crystal child. Original articles from' |
| | **Completion:** | ' our library related to the How To Recognize A Crystal Child. See Table of Contents for further available material (downloadable resources' |
| 3 | **Prompt:** | 'Wampsville, New York —Wampsville, New York— —● Total——1.0 sq mi' |
| | **Completion:** | ' (2.6 km2)— —● Land——1.0 sq mi (2.6 km2)— ' |
| 4 | **Prompt:** | 'Report on Stromboli (Italy) — 12 March-18 March 2003 Smithsonian / US Geological Survey Weekly Vol' |
| | **Completion:** | 'canic Activity Report, 12 March-18 March 2003 Managing Editor: Gari Mayberry Please cite this report' |
| 5 | **Prompt:** | 'Presentation on theme: "The Great (gym) Divide Curricula by Design #3 M. Fischer."—' |
| | **Completion:** | ' Presentation transcript: The Great (gym) Divide Curricula by Design #3 M. Fischer The Great' |
| 6 | **Prompt:** | 'Course Hero. "Lord of the Flies Study Guide." Course Hero. 15 Sep. 2016. Web. 29 May 20' |
| | **Completion:** | '23. ¡https://www.coursehero.com/lit/Lord-of-the-Flies/¿. ' |
| 7 | **Prompt:** | 'Manada Gap, Pennsylvania facts for kids Quick facts for kids Manada Gap, Pennsylvania —Time zone——UTC' |
| | **Completion:** | '-5 (Eastern (EST))— —● Summer (DST)——UTC-4 (EDT)— ' |
| 8 | **Prompt:** | 'Scale Zoology Cosmoid Scales A selection of articles related to scale zoology cosmoid scales. Original' |
| | **Completion:** | ' articles from our library related to the Scale Zoology Cosmoid Scales. See Table of Contents for further available material (' |
| 9 | **Prompt:** | 'Atomic Nucleus History A selection of articles related to atomic nucleus history. Original articles from our library related to' |
| | **Completion:** | ' the Atomic Nucleus History. See Table of Contents for further available material (downloadable resources) on Atomic Nucleus' |
| 10 | **Prompt:** | 'Mangoverde :: World Bird Guide :: Pheasants and Partridges :: Common Quail Common Quail Cot' |
| | **Completion:** | 'urnix coturnix Described by: Linnaeus (1758) Alternate common name(s' |
| 11 | **Prompt:** | 'Set Builder Notation Variations A selection of articles related to set builder notation variations. Original articles from our library related' |
| | **Completion:** | ' to the Set Builder Notation Variations. See Table of Contents for further available material (downloadable resources) on Set Builder' |
| 12 | **Prompt:** | '—1648 by topic— —Arts and science— —Birth and death categories— —Births –' |
| | **Completion:** | ' Deaths— —Establishments and disestablishments categories— —Establishments – Disestablishments— —' |
| 13 | **Prompt:** | 'Tamil Script The Tamil Letters A selection of articles related to tamil script the tamil letters. Original articles from' |
| | **Completion:** | ' our library related to the Tamil Script The Tamil Letters. See Table of Contents for further available material (downloadable resources) on' |
| 14 | **Prompt:** | 'Manuel I of PortugalFrom Wikipedia, the free encyclopediaJump to: navigation, search This article does not cite any references or' |
| | **Completion:** | ' sources. Please help improve this article by adding citations to reliable sources. Unsourced material may be challenged and removed. (' |
| 15 | **Prompt:** | 'Image 1 of 12 Image 2 of 12 Image 3 of 12 Image 4 of 12 Image 5 of 12 ' |
| | **Completion:** | 'Image 6 of 12 Image 7 of 12 Image 8 of 12 Image 9 of 12 Image 10 of 12 ' |
| 16 | **Prompt:** | '—Nutritional Guidelines (per serving)— —Servings: 3 pint jars (96 servings)— —Amount per serving' |
| | **Completion:** | '— —% Daily Value*— —Total Fat 0g——0%— —Saturated Fat 0g' |
| 17 | **Prompt:** | 'Presentation on theme: "Corpus Linguistics and Stylistics PALA Summer School, Maribor, 2014' |
| | **Completion:** | '."— Presentation transcript: Corpus Linguistics and Stylistics PALA Summer School, Maribor,' |
| 18 | **Prompt:** | 'Presentation on theme: "Lunar Research Station Design Submitted by West Valley Elementary GATE Team October 31, 2006' |
| | **Completion:** | '."— Presentation transcript: Lunar Research Station Design Submitted by West Valley Elementary GATE Team October 31,' |
| 19 | **Prompt:** | 'Some daily events in the changing sky for February 19 27. Friday, February 19 Saturday, February 20 Sunday,' |
| | **Completion:** | ' February 21 Monday, February 22 Tuesday, February 23 Wednesday, February 24 Thursday, February 25 Friday,' |

Table 6: **Randomly sampled lingering sequences at filtering strength** $n = 10$ **filter**. Sequence length $k = 50$.

| Idx | Lingering sequences at filtering strength $n = 10$ **filter**. | |
|---|---|---|
| 0 | **Prompt:** | 'Presentation on theme: "HELPING YOUR CHILD WITH NUMERACY: ADDITION AND SUBTRACTION."' |
| | **Completion:** | '— Presentation transcript: HELPING YOUR CHILD WITH NUMERACY: ADDITION AND SUBTRACTION ' |
| 1 | **Prompt:** | '—Wednesday——2:00 PM - 3:40 PM——lesson——Lecture Hall 1.2— ' |
| | **Completion:** | '—Thursday——2:00 PM - 3:40 PM——lesson——Lecture Hall 1.2— ' |
| 2 | **Prompt:** | 'How to define the cosine ratio and identify the cosine of an angle in a right triangle. How to define the' |
| | **Completion:** | ' sine ratio and identify the sine of an angle in a right triangle. How to define the tangent ratio and' |
| 3 | **Prompt:** | 'Q1. A series is given with one term missing. Select the correct alternative from the given ones that will complete the series' |
| | **Completion:** | '. Q2. A series is given with one term missing. Select the correct alternative from the given ones that will complete' |
| 4 | **Prompt:** | 'History of False Teeth Length: 497 words (1.4 double-spaced pages) - - -' |
| | **Completion:** | ' - - - - - - - - - - - - - - - - - - - - - - -' |
| 5 | **Prompt:** | 'Presentation on theme: "Yoghurt!!! Find the dairy cow on each page!!! By Daisy Mason and Brigette Roberts' |
| | **Completion:** | '."— Presentation transcript: Yoghurt!!! Find the dairy cow on each page!!! By Daisy Mason and Brigette' |
| 6 | **Prompt:** | 'Protecting People with Disabilities in the Ebbs and Flows of the COVID-19 Pandemic Protecting People' |
| | **Completion:** | ' with Disabilities in the Ebbs and Flows of the COVID-19 Pandemic The COVID-19 pand' |
| 7 | **Prompt:** | 'Presentation on theme: "Aceh Poverty Assessment The impact of the Conflict, the Tsunami and Reconstruction on Poverty' |
| | **Completion:** | ' in Aceh."— Presentation transcript: Aceh Poverty Assessment The impact of the Conflict, the Tsunami' |
| 8 | **Prompt:** | 'Presentation on theme: "THE MIX-AERATOR Innovation In Pond & Lagoon Aeration & Mixing."' |
| | **Completion:** | '— Presentation transcript: THE MIX-AERATOR Innovation In Pond & Lagoon Aeration & Mixing ' |
| 9 | **Prompt:** | 'Some daily events in the changing sky for February 8 16. Friday, February 8 Saturday, February 9 Sunday,' |
| | **Completion:** | ' February 10 Monday, February 11 Tuesday, February 12 Wednesday, February 13 Thursday, February 14 Friday,' |
| 10 | **Prompt:** | 'Essays on mercutio Romeo and mercutio essays: over 180,000 romeo and merc' |
| | **Completion:** | 'utio essays, romeo and mercutio term papers, romeo and mercutio research paper, book' |
| 11 | **Prompt:** | 'Presentation on theme: "Fabric Construction Fashion Design, Textiles & Merchandising Mrs. Moscinski."—' |
| | **Completion:** | ' Presentation transcript: Fabric Construction Fashion Design, Textiles & Merchandising Mrs. Moscinski Fabric' |
| 12 | **Prompt:** | 'Presentation on theme: "Chapter 4 - Building Compassionate School-Community Partnerships That Work Chapter 4 - Building Comp' |
| | **Completion:** | 'assionate School-Community Partnerships That Work."— Presentation transcript: Chapter 4 - Building Compassionate School-' |
| 13 | **Prompt:** | 'Tracing Names: Letter AA — B — C — D — E — F — G — H — I — J —' |
| | **Completion:** | ' K — L — M — N — O — P — Q — R — S — T — U — V — W' |
| 14 | **Prompt:** | 'Dictionary of Financial, Economic, and Business Terms A — B — C — D — E — F — G —' |
| | **Completion:** | ' H — I — J — K — L — M — N — O — P — Q — R — S — T' |
| 15 | **Prompt:** | 'Canons of the Seven Ecumenical Councils. The First Ecumenical Council. Second Ecumenical' |
| | **Completion:** | ' Council. Third Ecumenical Council. Fourth Ecumenical Council. Fifth Ecumenical Council. ' |
| 16 | **Prompt:** | 'Welsh Levels of Care E-Learning Program Glossary Special — A — B — C — D — E — F' |
| | **Completion:** | ' — G — H — I — J — K — L — M — N — O — P — Q — R —' |
| 17 | **Prompt:** | 'Presentation on theme: "Tap Water Intrusion Effects on Microbial Life Anthony DeRenzo Grade 10 Pittsburgh Central Catholic' |
| | **Completion:** | ' High School."— Presentation transcript: Tap Water Intrusion Effects on Microbial Life Anthony DeRenzo Grade 10' |
| 18 | **Prompt:** | 'Presentation on theme: "Dr. Anand Srinivasan for MBBS 2013 on 10/10/2013."' |
| | **Completion:** | '— Presentation transcript: Dr. Anand Srinivasan for MBBS 2013 on 10/10/2013 ' |
| 19 | **Prompt:** | 'Presentation on theme: "Ashok Sinha O/o the Director General (Audit) Central, Chandigar' |
| | **Completion:** | 'h."— Presentation transcript: Ashok Sinha O/o the Director General (Audit) Central, Chand' |

Table 7: **Randomly sampled lingering sequences at filtering strength** $n = 5$ **filter**. Sequence length $k = 50$.

| Idx | Lingering sequences at filtering strength $n = 5$ **filter**. | |
|---|---|---|
| 0 | **Prompt:** | 'Water, sanitation and hygiene: the foundation for building resilience in climate-vulnerable communities - Water, sanitation and hygiene:' |
| | **Completion:** | ' the foundation for building resilience in climate-vulnerable communities - Water, sanitation and hygiene: the foundation for building resilience in' |
| 1 | **Prompt:** | '- 1 What is Adrenoleukodystrophy disease? - 2 Adrenoleukodystrophy Causes - 3' |
| | **Completion:** | ' Adrenoleukodystrophy Symptoms - 4 Adrenoleukodystrophy Diagnosis - 5 Adrenoleukody' |
| 2 | **Prompt:** | 'MATH105 April 2017 ● Q1 (a) ● Q1 (b) ● Q1 (c) ●' |
| | **Completion:** | ' Q1 (d) ● Q1 (e) ● Q1 (f) ● Q1 (g) ● Q' |
| 3 | **Prompt:** | 'Native to North America STATE DISTRIBUTION (USDA): AL, AR, CT, DC, DE, FL,' |
| | **Completion:** | ' GA, IA, IL, IN, KS, KY, LA, MA, MD, ME, MI, MN, MO' |
| 4 | **Prompt:** | 'What are the 7 notes of a major scale? The scale degrees are: - 1st: Tonic. ' |
| | **Completion:** | '- 2nd: Supertonic. - 3rd: Mediant. - 4th: Subdominant.' |
| 5 | **Prompt:** | 'Ten Times Table And Random Test Lyrics 10 x 1 = 10 10 x 2 = 20 10 x 3 =' |
| | **Completion:** | ' 30 10 x 4 = 40 10 x 5 = 50 10 x 6 = 60 10 x 7 = 70' |
| 6 | **Prompt:** | 'Accuracy Of Data 914 words (2.6 double-spaced pages) - - - - - -' |
| | **Completion:** | ' - - - - - - - - - - - - - - - - - - - - - - -' |
| 7 | **Prompt:** | 'Print Texting RULES! Reading Comprehension with Fourth Grade Work Print Texting RULES! Reading Comp' |
| | **Completion:** | 'rehension with Fifth Grade Work Print Texting RULES! Reading Comprehension with Sixth Grade Work Print Text' |
| 8 | **Prompt:** | 'Glossary of Legal Terms A - B - C - D - E - F - G - H' |
| | **Completion:** | ' - I - J - K - L - M - N - O - P - Q - R' |
| 9 | **Prompt:** | 'Chef is at x=0. 1-jump: he will move from x -¿ x + 1 2-' |
| | **Completion:** | 'jump: he will move from x -¿ x + 2 3-jump: he will move from x -¿ x + 3' |
| 10 | **Prompt:** | 'Acting Minister of the Environment, Denis Kellman (centre, ', ', ', '' |
| | **Completion:** | '', ', ', ', ', ', ', ', ', '' |
| 11 | **Prompt:** | 'MI Science Standards Special — A — B — C — D — E — F — G — H — I — J' |
| | **Completion:** | ' — K — L — M — N — O — P — Q — R — S — T — U — V —' |
| 12 | **Prompt:** | 'Collective Nouns for Birds —Pages:——A,——B,——C,——D,——E' |
| | **Completion:** | ',——F,——G,——H,——I,——J,——K,——L,——M,' |
| 13 | **Prompt:** | '7 Wicked Winter Health Myths By: Laura Roberson - Winter Health Myth # 1 - Winter Health Myth #' |
| | **Completion:** | ' 2 - Winter Health Myth # 3 - Winter Health Myth # 4 - Winter Health Myth # 5 - Winter' |

| Idx | Lingering sequences at filtering strength $n = 5$ filter. | |
|---|---|---|
| 14 | **Prompt:** | 'HISTORY CRIME AND PUNISHMENT HISTORY CRIME AND PUNISHMENT HISTORY CRIME AND' |
| | **Completion:** | ' PUNISHMENT HISTORY CRIME AND PUNISHMENT HISTORY CRIME AND PUNISHMENT ' |
| 15 | **Prompt:** | 'Define Gyromitra infula. Gyromitra infula synonyms, Gyromitra in' |
| | **Completion:** | 'fula pronunciation, Gyromitra infula translation, English dictionary definition of Gyromitra infula.' |
| 16 | **Prompt:** | 'Some daily events in the changing sky for December 19 27. Friday, December 19 Saturday, December 20 Sunday,' |
| | **Completion:** | ' December 21 Monday, December 22 Tuesday, December 23 Wednesday, December 24 Thursday, December 25 Friday,' |
| 17 | **Prompt:** | 'Length: 1122 words (3.2 double-spaced pages) - - - - - - - - -' |
| | **Completion:** | ' - - - - - - - - - - - - - - - - - - - - - - - -' |
| 18 | **Prompt:** | 'Letter E Names: Page 2A — B — C — D — E — F — G — H — I — J' |
| | **Completion:** | ' — K — L — M — N — O — P — Q — R — S — T — U — V —' |
| 19 | **Prompt:** | 'A Complete Illustrated History of Robots in the Movies (chronological by film title) Intro — Part 1 —' |
| | **Completion:** | ' Part 2 — Part 3 — Part 4 — Part 5 — Part 6 — Part 7 — Part 8 — Part 9 — Part' |
| 20 | **Prompt:** | '1. George Washington - Term of Office (1789-1797) 2. John Adams (1797-18' |
| | **Completion:** | '01) 3. Thomas Jefferson (1801-1809) 4. James Madison (1809-1817' |

## A.3. Visualizing neighbors of lingering sequences $\mathcal{D}_{\text{linger}}^{(n)}$ in the pre-training set

A key aspect worth studying for lingering sequences is what contributed to their existence. We randomly sample 2 lingering sequences for filtering strengths $n = \{50, 20, 10\}$, and perform a very costly search of Levenshtein edit-distance neighbor search: perform a sliding window over the pre-training tokens, and check the edit distance of each window to the lingering sequence. By construction, lingering sequences have already been removed from the training set; the hope is thus to identify and visualize *neighboring* sequences that may have led to these lingering sequences.

Fig. 8 visualizes these randomly sampled lingering sequences, and two randomly sampled neighbors (with edit distance < 20 tokens) for each of them. We also visualize the histogram of these neighbors at different distances. We note that:

- These visualizations suggest that lingering sequences are very likely the result of either *near-duplicate* training data and/or the generalization capabilities of LLMs.
- **By observing the neighboring sequences, we gain insights into how we may *adversarially* game the $n$-gram membership definition.** For example, observing the second sequence in Fig. 8 can provide intuition for the *chunking* (§5.1) method we presented in §5 to adversarially construct fine-tuning sequences that avoid $n$-gram overlap, and observing the second last can provide intuition for the *token dropout* (§5.2) method.

**Lingering Seq (n = 50 filter):** The Sixth Amendment to the U.S. Constitution reads, "In all criminal prosecutions, the accused shall enjoy the right to a speedy and public trial, by an impartial jury of the State and district wherein the crime shall have been committed, which

**Neighbor #1:** .\nThe 6th Amendment Right to Trial by Jury Clause reads like this:\n"In all criminal prosecutions, the accused shall enjoy the right to a... trial, by an impartial jury of the State and district where in the crime shall have been committed

**Neighbor #2:** nor shall property be taken for public, without just compensation.\n- Amendment VI In all criminal prosecutions the accused shall enjoy the right to a speedy and public trial, by an impartial jury of the state and district wherein the crime shall have been committed

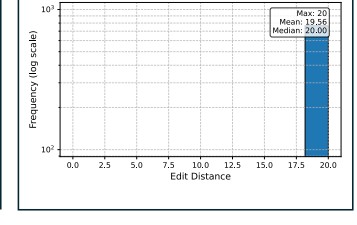

**Lingering Seq (n = 50 filter):** If you want to pay for essay for unique writing Looking At The Chinese Lifestyle And Norms, just click Order button. We will write a custom essay on Looking At The Chinese Lifestyle And Norms specifically for you!

**Neighbor #1:** If you want to pay for essay for unique writing The role of cybersecurity and cybercrime, just click Order button. We will write a custom essay on The role of cybersecurity and cybercrime specifically for you!

**Neighbor #2:** If you want to pay for essay for unique writing Gender Roles and Lady Macbeth, just click Order button. We will write a custom essay on Gender Roles and Lady Macbeth specifically for you!

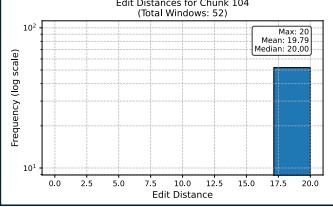

**Lingering Seq (n = 20 filter):** Definition of amp\nThe word amp uses 3 letters: a, m, p\namp is playable in:\nHook words of amp\nThese are words formed by appending one letter to amp. Extend an already existing word on the board.

**Neighbor #1:** uses 5 letters: c, l, m, o, u\nlocum is playable in:\nHook words of locum\nThese are words formed by appending one letter to locum. Extend an already existing word on the board.

**Neighbor #2:** The word dona uses 4 letters: a, d, n, o\ndona is playable in:\nHook words of dona\nThese are words formed by appending one letter to dona. Extend an already existing word on

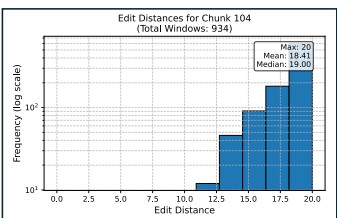

**Lingering Seq (n = 20 filter):** A selection of articles related to kulin brahmins.\nOriginal articles from our library related to the Kulin Brahmins. See Table of Contents for further available material (downloadable resources) on Kulin Brahmins.\n- The

**Neighbor #1:** selection of articles related to the creation of adam.\nOriginal articles from our library related to the The Creation Of Adam. See Table of Contents for further available material (downloadable resources) on The Creation Of Adam.\n- The Aeonic Perspective of

**Neighbor #2:** .<|endoftext|>A selection of articles related to sufi texts.\nOriginal articles from our library related to the Sufi Texts. See Table of Contents for further available material (downloadable resources) on Sufi Texts.\n- Select Cross-

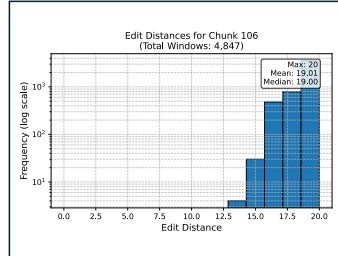

**Lingering Seq (n = 10 filter):** 2005 AMC 10A Problems\n- 1 Problem 1\n- 2 Problem 2\n- 3 Problem 3\n- 4 Problem 4\n- 5 Problem 5\n- 6 Problem 6\n- 7 Problem 7\n- 8 Problem 8\n- 9 Problem 9

**Neighbor #1:** \n- 1 Article 1\n- 2 Article 2\n- 3 Article 3\n- 4 Article 4\n- 5 Article 5\n- 6 Article 6\n- 7 Article 7\n- 8 Article 8\n- 9 Article 9\n- 10 Article 10

**Neighbor #2:** Have fun. Don't die.\n- 1 Problem 13\n- 2 Problem 14\n- 3 Problem 13\n- 4 Problem 14\n- 5 Problem 13\n- 6 Problem 14\n- 7 Problem 14\n- 8 Problem 13\n- 9

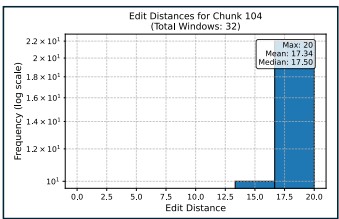

**Lingering Seq (n = 10 filter):** Little Big Store in Raymond\nLocation of Raymond, Mississippi\n|• Total||3.0 sq mi (7.7 km2)|\n|• Land||3.0 sq mi (7.7 km2)|\n|• Water||

**Neighbor #1:** \nCA-26: Julia Brownley (D)\n|• Total||32.25 sq mi (83.53 km2)|\n|• Land||21.82 sq mi (56.50 km2)|\n|• Water||

**Neighbor #2:** in Androscoggin County and the state of Maine\n|• Total||62.72 sq mi (162.44 km2)|\n|• Land||59.26 sq mi (153.48 km2)|\n|• Water||

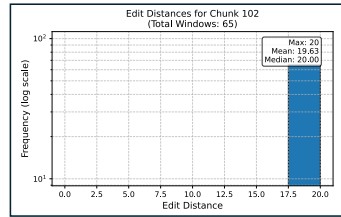

Figure 8: Visualizing the neighbors (Levenshtein edit distance < 20 tokens) of a few randomly selected lingering sequence from $\mathcal{D}_{\text{linger}}^{(10)}$, $\mathcal{D}_{\text{linger}}^{(20)}$, and $\mathcal{D}_{\text{linger}}^{(50)}$.

## A.4. Model architectures

Our 350M, 774M, and 1.6B-parameter model architectures follow directly from the original GPT-2 paper (Radford et al., 2019) and LLM.c implementation (Karpathy, 2024). We create a custom architecture for 2.8B parameters by adjusting the number of layers, channels, and attention headas in the model, again following Karpathy (2024). The model configurations can be found in Table 8.

Table 8: Configurations of models (GPT-2 architecture) at different sizes for our pre-training experiments.

| Parameters | 304M | 774M | 1.6B | 2.8B |
|---|---|---|---|---|
| Layers | 24 | 36 | 48 | 60 |
| Channels | 1024 | 1280 | 1600 | 1920 |
| Attention heads | 16 | 20 | 26 | 30 |

## A.5. Training hyperparameters

We follow the pre-training configurations outlined in (Karpathy, 2024), particularly `https://github.com/karpathy/llm.c/discussions/677`. Table 9 summarizes the configurations.

Table 9: Training configurations for pre-training experiments.

| Hyperparameter | Value |
|---|---|
| # Training Tokens | 33.6 billion |
| Compute | 8 NVIDIA H100 days (1.6B parameter model) |
| Micro-Batch Size | 16 |
| Max Sequence Length | 1024 |
| Total Batch Size | $2^{20} = 1,048,576$ tokens |
| Gradient Accumulation Steps | 8 |
| Weight Decay | 0.1 |
| Learning Rate | 6e-4 |
| LR Schedule | Cosine |
| LR Decay | decay to 10% of max LR |
| Warmup Iterations | 700 iterations |
| Total Training Steps | 32,000 |

## A.6. Tabled results of Fig. 3

We additionally provide the table version of Fig. 3 in Table 10 (for default sequence length 50).

Table 10: (Table version of Fig. 3) **Amount of lingering sequences** as fraction of $|\mathcal{D}_{\text{mem}}|$ across model sizes and filtering strengths (values of $n$-gram filter). Sequence length $k = 50$. Recall setup in Section 4.

| Model Size | $|\mathcal{D}_{\text{mem}}|$ | Filtering Strength | | | |
|---|---|---|---|---|---|
| | | $n = 5$ | $n = 10$ | $n = 20$ | $n = 50$ |
| 304M | 76648 | 0.0175 | 0.0402 | 0.0978 | 0.4793 |
| 774M | 116270 | 0.0132 | 0.0300 | 0.0916 | 0.4013 |
| 1.6B | 151598 | 0.0109 | 0.0259 | 0.0796 | 0.4007 |
| 2.8B | 175813 | 0.0099 | 0.0247 | 0.0869 | 0.4068 |

## A.7. Additional Results: Longer Sequence Length

We also experiment with a longer sequence length of $k = 100$ (i.e., 50-token prompt, 50-token completion). In general, we made similar observations in terms of both the amount and nature of the lingering sequences. Table 11 shows the fraction of lingering sequences in terms of the identified memorized sequences $\mathcal{D}_{\mathrm{mem}}$. We provide visualizations of these sequences in Table 12 (exact filtering, $n = 100$) and Table 12 (strong filtering, $n = 10$), and note that the content of the lingering sequences are similar to those with sequence length $k = 50$.

Table 11: Amount of lingering sequences as a fraction of the identified memorized sequences $|\mathcal{D}_{\mathrm{mem}}|$ across filtering strengths (values of $n$-gram filter). Sequence length $k = 100$. Recall setup in Section 4; compare to Table 10.

| Model Size | $|\mathcal{D}_{\mathbf{mem}}|$ | Filtering Strength | | | |
|---|---|---|---|---|---|
| | | $n = 10$ | $n = 20$ | $n = 50$ | $n = 100$ |
| 1.6B | 88803 | 0.0109 | 0.0259 | 0.0796 | 0.4007 |

Table 12: **Randomly sampled lingering sequences at filtering strength** $n = 100$ **(exact) filter**. Sequence length $k = 100$.

| Idx | Lingering sequences at filtering strength $n = 100$ **(exact) filter.** |
|---|---|
| 0 | **Prompt:** "Course Hero. "Romeo and Juliet Study Guide." Course Hero. 28 July 2016. Web. 16 Jan. 2019. ¡https://www.coursehero.com/lit/Romeo-and-Juliet/¿. Course" 
 **Completion:** " Hero.  (2016, July 28).  Romeo and Juliet Study Guide.  In Course Hero.  Retrieved January 16, 2019, from https://www.coursehero.com/lit/Romeo-and-Juliet/ (Course Hero," |
| 1 | **Prompt:** "ESQUIEN INDIANS. The Esquien Indians were a Karankawan group that entered Nuestra Señora de la Candelaria Mission on the San Gabriel River in 1750. Frederick Webb Hodge, ed.," 
 **Completion:** " Handbook of American Indians North of Mexico (2 vols., Washington: GPO, 1907, 1910; rpt., New York: Pageant, 1959). The following, adapted from the Chicago Manual of Style, 15th edition, is the" |
| 2 | **Prompt:** "See what questions a doctor would ask. During a consultation, your doctor will use various techniques to assess the symptom: Heel pain. These will include a physical examination and possibly diagnostic tests. (Note: A physical exam is always done," 
 **Completion:** " diagnostic tests may or may not be performed depending on the suspected condition) Your doctor will ask several questions when assessing your condition. It is important to openly share any pertinent information to help your doctor make an accurate diagnosis. It is also very important to" |
| 3 | **Prompt:** "—Product #: EMC0775025_TQ— A Is for Apple (Resource Book Only) eBookGrade 2—Grade 3—Grade 4—Grade 5 Please Note: This ebook is a digital download, NOT a physical product" 
 **Completion:** ". After purchase, you will be provided a one time link to download ebooks to your computer. Orders paid by PayPal require up to 8 business hours to verify payment and release electronic media. For immediate downloads, payment with credit card is required. " |
| 4 | **Prompt:** "Family History and Genealogy Resources by Surname Haycock Surname Origin A name probably given to a foundling exposed in a hayfield. Source: An Etymological Dictionary of Family and Christian Names With an Essay" 
 **Completion:** " on their Derivation and Import; Arthur, William, M.A.; New York, NY: Sheldon, Blake, Bleeker & CO., 1857. Haycock Surname Meaning and Family Facts There is more to Haycock family" |
| 5 | **Prompt:** "'Lake Uvs and its surrounding wetlands - Site number:1379 - Area:585,000 ha - Designation date:22-03-2004 - Coordinates:50° 19'N 92° 45'E Materials'" 
 **Completion:** " presented on this website, particularly maps and territorial information, are as-is and as-available based on available data and do not imply the expression of any opinion whatsoever on the part of the Secretariat of the Ramsar Convention concerning the legal status of" |
| 6 | **Prompt:** "Barrier Methods of Birth Control (cont.) Melissa Conrad Stöppler, MD Melissa Conrad Stöppler, MD, is a U.S. board-certified Anatomic Pathologist with subspecialty training in" 
 **Completion:** "' the fields of Experimental and Molecular Pathology. Dr. Stöppler's educational background includes a BA with Highest Distinction from the University of Virginia and an MD from the University of North Carolina. She completed residency training in Anatomic Pathology at'" |
| 7 | **Prompt:** "Definition of Japanese deer 1. Noun. Small deer of Japan with slightly forked antlers. Generic synonyms: Cervid, Deer Group relationships: Cervus, Genus Cervus Japanese Deer Pictures Click" 
 **Completion:** " the following link to bring up a new window with an automated collection of images related to the term: Japanese Deer Images Lexicographical Neighbors of Japanese Deer Literary usage of Japanese deer Below you will find example usage of this term as" |
| 8 | **Prompt:** "Comparing Fractions (G) In this comparing fractions practice worksheet, 5th graders examine 10 pairs of fractions. Students identify each of the pairs of fractions as greater than, less than, or equal to one another. 3 Views" 
 **Completion:** " 0 Downloads Fraction Equivalence, Ordering, and Operations Need a unit to teach fractions to fourth graders? Look no further than this well-developed and thorough set of lessons that takes teachers through all steps of planning, implementing," |
| 9 | **Prompt:** "PREAMBLEWhereas recognition of the inherent dignity and of the equal and inalienable rights of all members of the human family is the foundation of freedom, justice and peace in the world, Whereas disregard and contempt for human rights have resulted in barbar" 
 **Completion:** "ous acts which have outraged the conscience of mankind, and the advent of a world in which human beings shall enjoy freedom of speech and belief and freedom from fear and want has been proclaimed as the highest aspiration of the common people, Whereas it is essential" |

Table 13: **Randomly sampled lingering sequences at filtering strength** $n = 10$ **filter**. Sequence length $k = 100$.

| Idx | Lingering sequences at filtering strength $n = 10$ **filter.** |
|---|---|
| 0 | **Prompt:** "Presentation on theme: "REVIEW We can tell how many electrons and atom will gain or lose by looking at its valence. Metals like to lose electrons. (Cations) –Ex. Na + Nonmetals."— Presentation" 
 **Completion:** " transcript: REVIEW We can tell how many electrons and atom will gain or lose by looking at its valence. Metals like to lose electrons. (Cations) –Ex. Na + Nonmetals like to gain electrons. (An" |
| 1 | **Prompt:** "—I ● II ● III ● IV ● V ● VI ● VII ● VIII ● IX ● X ● XI ● XII ● XIII ● XIV ● Schedule— - 1 Features - 2 Preamble - 3 Article I - 4 Article" 
 **Completion:** " II - 5 Article III - 6 Article IV - 7 Article V - 8 Article VI - 9 Article VII - 10 Article VIII - 11 Article IX - 12 Article X - 13 Article XI - 14 Article" |

| Idx | Lingering sequences at filtering strength $n = 10$ filter. |
|---|---|
| 2 | **Prompt:** "ENGL 301 Course Introduction this course will be an independent study of a topic of the student's choice in English literature and film. ENGL 302 Course Introduction for ENGL 302 – Independent Study In English (ENGL 302) this" 
 **Completion:** " course will be an independent study of a topic of the student's choice in English literature and film. ENGL 303 Course Introduction for ENGL 303 – Independent Study In English (ENGL 303) this course will be an independent study of" |
| 3 | **Prompt:** "'NRL's MISSE-8 Launched Aboard STS-134 - About NRL - Doing Business - Public Affairs & Media - Public Affairs Office - News Releases - 2017 News Releases - 2016 News Releases -'" 
 **Completion:** " 2015 News Releases - 2014 News Releases - 2013 News Releases - 2012 News Releases - 2011 News Releases - 2010 News Releases - 2009 News Releases - 2008 News Releases - 2007 News Releases - 2006 News Releases -" |
| 4 | **Prompt:** "Early Movie Inventions Early Movie Exhibitions The First Movie Posters Movies and Movie Posters of the 1900's Movies and Movie Posters of the 1910's Movies and Movie Posters of" 
 **Completion:** " the 1920's Movies and Movie Posters of the 1930's Movies and Movie Posters of the 1940's Movies and Movie Posters of the 1950's Movies and Movie Post" |
| 5 | **Prompt:** "Presentation on theme: "U.S.A- 1865-1918 Expansion westward after civil war (1861-1865) Gold, silver, and land Natives were pushed aside & subdued. Railways were key to Western."" 
 **Completion:** "— Presentation transcript: U.S.A- 1865-1918 Expansion westward after civil war (1861-1865) Gold, silver, and land Natives were pushed aside & subdued. Railways were key to Western expansion" |
| 6 | **Prompt:** "FILTER BY Year: - 2014 http://pub2web.metastore.ingenta.com/ns/yearOfPublication 2014 - 2013 http://pub2web.metastore.ingenta.com/" 
 **Completion:** "ns/yearOfPublication 2013 - 2012 http://pub2web.metastore.ingenta.com/ns/yearOfPublication 2012 - 2011 http://pub2web.metastore.ingenta." |
| 7 | **Prompt:** "Please note that the content of this book primarily consists of articles available from Wikipedia or other free sources online. Pages: 181. Not illustrated. Chapters: 11th-Century Roman Catholic Church Councils, 12th-Century Roman Catholic Church Council" 
 **Completion:** "s, 13th-Century Roman Catholic Church Councils, 14th-Century Roman Catholic Church Councils, 15th-Century Roman Catholic Church Councils, 16th-Century Roman Catholic Church Councils, 17th-Cent" |
| 8 | **Prompt:** "Presentation on theme: "1 Chapter 10 Graphene-based Nanocomposites 10.1 Introduction of composites 10.2 Introduction of graphene-polymer nanocomposites 10.3 Processing of graphene-polymer."— Present" 
 **Completion:** "ation transcript: 1 Chapter 10 Graphene-based Nanocomposites 10.1 Introduction of composites 10.2 Introduction of graphene-polymer nanocomposites 10.3 Processing of graphene-polymer nanocomposites 10" |
| 9 | **Prompt:** "Bentham is known by most as the father of utilitarianism. He wrote in favor of free-markets, a pragmatic view of rights, and rational policy-making. - A Apply A filter - B Apply B filter - C" 
 **Completion:** " Apply C filter - D Apply D filter - E Apply E filter - F Apply F filter - G Apply G filter - H Apply H filter - I Apply I filter - J Apply J filter - K Apply K" |

## A.8. Additional Results: Almost-Lingering Sequences

When reporting the fraction of lingering sequences (Table 10 and Table 3), the lingering sequences are defined in terms of *exact* completion (Def. 3.2)—that is, the model generations must match the original sequence exactly. It is then natural to also ask whether there would be any *almost*-lingering sequences—or sequence completions that are at small edit distance from the original sequences (as in Def. 3.3).

To check for almost-lingering sequences, we can perform the same completion check procedure against the set of identified sequences $\mathcal{D}_{\mathrm{mem}}$ as before (recall Section 4.1), except now we also add sequences that are only a few tokens ($< 20$ off).

Table 14 shows the amount of almost-lingering sequences for sequence length $k = 50$, across different edit distance thresholds $\{0, 5, 10, 20\}$ (where 0 corresponds to exact lingering sequences; see Table 10). Similarly, Table 15 shows the statistics for sequence length $k = 100$. We make the following observations:

- For strong filtering (low $n$-gram filter), there are relatively few such almost-lingering sequences; *e.g.*, at $n = 5$, the fraction grows from $\approx 1\%$ (Fig. 3) to up to around $\approx 2\%$ within edit distance of 10 tokens (Table 14).
- For loose filtering (large $n$-gram filter), the fraction grows more substantially; *e.g.*, at $n = 50$, the lingering fraction grows from $\approx 40\%$ to $\approx 60\%$ within a distance of 10 for sequence length $k = 50$.

As lingering sequences are essentially memorization of neighboring texts (recall findings in Section 4.2) at weak filters (small values $n$), this suggests that data removal needs to carefully consider the definition of when two sequences are (approximately) equal.

Table 14: **Almost-lingering sequences** (sequence length $k = 50$) as fraction of $|\mathcal{D}_{\mathrm{mem}}| = 151598$ (for 1.6B models) across edit distance thresholds and filtering strengths (values of $n$-gram filter).

| Edit Distance | Filtering Strength | | | |
|:---:|:---:|:---:|:---:|:---:|
| | $n = 5$ | $n = 10$ | $n = 20$ | $n = 50$ |
| 0 | 0.0109 | 0.0259 | 0.0796 | 0.4007 |
| 5 | 0.0140 | 0.0365 | 0.1486 | 0.5386 |
| 10 | 0.0217 | 0.0482 | 0.2051 | 0.6105 |
| 20 | 0.1571 | 0.2227 | 0.4114 | 0.7638 |

Table 15: **Almost-lingering sequences** (sequence length $k = 100$) as fraction of $|\mathcal{D}_{\mathrm{mem}}| = 88803$ (for 1.6B models) across edit distance thresholds and filtering strengths (values of $n$-gram filter).

| Edit Distance | Filtering Strength | | | |
|:---:|:---:|:---:|:---:|:---:|
| | $n = 10$ | $n = 20$ | $n = 50$ | $n = 100$ |
| 0 | 0.0082 | 0.0216 | 0.1418 | 0.4335 |
| 5 | 0.0097 | 0.0267 | 0.2344 | 0.5332 |
| 10 | 0.0111 | 0.0305 | 0.2837 | 0.6034 |
| 20 | 0.0168 | 0.0599 | 0.3249 | 0.6670 |

### A.9. Additional Results: Persistence and Overlap of Lingering Sequences Over Repeated Runs

We are also interested in exploring to what extent lingering sequences are due to random chance. We repeated the 1.6B pre-training run *from scratch* for five times for filtering strengths $n = \{10, 20, 50\}$.[4] Note that to perform a repeated run, we do *not* need to re-train the base model $\mathcal{M}_{\text{base}}$ and obtain a different set of memorized sequences $\mathcal{D}_{\text{mem}}$ (recall Section 4.1); it suffices to operate on the same $\mathcal{D}_{\text{mem}}$, and observe the variance of pre-training on the existence of lingering sequences $\mathcal{D}_{\text{linger}}^{(n)}$. This is because we are interested in studying the impact of training stochasticity on which sequences remain lingering sequences given a set of memorized sequences.

Table 16 presents the **persistence** of the lingering sequences $\mathcal{D}_{\text{linger}}^{(n)}$ over repeated runs, and Table 17 presents the **overlap** over repeated runs. Observe that:

- The amount of lingering sequences are fairly stable across runs (Table 16). This suggests that the existence of lingering sequences is heavily influenced by the nature of the pre-training dataset, more so than the randomness of the pre-training procedure (e.g., data shuffling, hardware randomness).
- Intriguingly, the overlap of lingering sequences across runs are not very high. That is, as we re-train $\mathcal{M}_{\text{filter}}^{(n)}$ multiple times on the same filtered data, we get a *different* set of lingering sequences, albeit having a similar number of them.[5] Nevertheless, as there are more repetitions, the intersection amount starts converging to a "core set" of lingering sequences (Table 17).

Table 16: **Persistence of lingering sequences over repeated pre-training runs** across filtering strengths (values of $n$-gram filter). Sequence length $k = 50$. Values are the number of sequences (and as fraction of $|\mathcal{D}_{\text{mem}}| = 151598$).

| Run Number | Filtering Strength | | | |
| --- | --- | --- | --- | --- |
| | $n = 5$ | $n = 10$ | $n = 20$ | $n = 50$ (exact filter) |
| 1 | 1652 (0.0109) | 3923 (0.0259) | 12066 (0.0796) | 60742 (0.4007) |
| 2 | - | 4085 (0.0269) | 12007 (0.0792) | 63297 (0.4175) |
| 3 | - | 4108 (0.0271) | 11684 (0.0771) | 61205 (0.4037) |
| 4 | - | 4086 (0.0270) | 12799 (0.0844) | 66077 (0.4359) |
| 5 | - | 4163 (0.0275) | 11590 (0.0765) | 60035 (0.3960) |
| mean $\pm$ std (absolute) | 1652 | 4073 $\pm$ 80.1723 | 12029.2 $\pm$ 425.7931 | 62271.2 $\pm$ 2191.4472 |
| mean $\pm$ std (fraction) | 0.0109 | 0.0269 $\pm$ 0.0005 | 0.0793 $\pm$ 0.0028 | 0.4108 $\pm$ 0.0145 |

Table 17: **Overlap of lingering sequences over repeated pre-training runs** across filtering strengths (values of $n$-gram filter). Sequence length $k = 50$. Values are the number of sequences (and as fraction of $|\mathcal{D}_{\text{mem}}| = 151598$).

| Intersection of $\mathcal{D}_{\text{linger}}^{(n)}$ from repeated runs | Filtering Strength | | |
| --- | --- | --- | --- |
| | $n = 10$ | $n = 20$ | $n = 50$ (exact filter) |
| 1 | 3923 (0.0259) | 12066 (0.0796) | 60742 (0.4007) |
| $1 \cap 2$ | 3330 (0.0220) | 8916 (0.0588) | 49606 (0.3272) |
| $1 \cap 2 \cap 3$ | 3034 (0.0200) | 7478 (0.0493) | 42512 (0.2804) |
| $1 \cap 2 \cap 3 \cap 4$ | 2822 (0.0186) | 6523 (0.0430) | 37654 (0.2484) |
| $1 \cap 2 \cap 3 \cap 4 \cap 5$ | 2666 (0.0176) | 5897 (0.0389) | 34185 (0.2255) |

---

[4]Due to the expensive nature of pre-training, we did not perform the repeated runs for $n = 5$ due to limited compute. We observed similar trends on the other filtering strengths.

[5]Here, for simplicity, we stick with set intersection to capture overlap, which implicitly performs exact string matching (as opposed to approximate matching).

## A.10. Prompt template for LLM-as-a-judge

Recall that in Section 4, we used LLM-as-a-judge to discern whether a lingering sequence is a generalizable pattern continuation, as opposed a verbatim memorization of a semantically useful text sequence (Fig. 5). Listing 1 illustrates the prompt template used for **Gemini 1.5 Pro** (Gemini Team et al., 2024) for classifying the lingering sequences.

Listing 1: Prompt template for LLM-as-a-judge for memorization vs. generalizable patterns (Fig. 5).

---

The following task involves determining whether the completions for a given prompt represent simple patterns, templates, or repetitive structures that could be generalized by a language model without requiring memorization. Examples of such simple patterns include sequences, repetitions, or other forms of predictable structures.

The response should only be one word: "yes" if the completions are simple patterns and "no" if they are not.

Examples of Classification:

Input:
− − −
idx=0 prompt:
'"6And so it was, that, while they were there, the days were accomplished that she should be delivered. 7'
model_name='M_gpt2_hf_base' completion:
'And she brought forth her firstborn son, and wrapped him in swaddling clothes, and laid him in a manger'
model_name='M_gpt2_hf_filter_k25' completion:
'And she brought forth her firstborn son, and wrapped him in swaddling clothes, and laid him in a manger'
model_name='openai−community/gpt2−xl' completion:
'And the angel of the Lord appeared to Joseph in a dream, saying, "Arise, take the young child and his'
all_match=False, fallback_match=False
− − −
Output: No

Input:
− − −
idx=1 prompt:
'"A wise and frugal government... shall restrain men from injuring one another, shall leave them otherwise free to regulate their'
model_name='M_gpt2_hf_base' completion:
' own pursuits of industry and improvement, and shall not take from the mouth of labor the bread it has earned. This is the'
model_name='M_gpt2_hf_filter_k25' completion:
' own pursuits of industry and improvement, and shall not take from the mouth of labor the bread it has earned. This is the'
model_name='openai−community/gpt2−xl' completion:
' own pursuits of industry and improvement, so long as they do not invade the equal rights of others."\n\nThe Constitution,'
all_match=False, fallback_match=False
− − −
Output: No

Input:
− − −
idx=2 prompt:
'"Aoccdrnig to a rscheearch at Cmabrigde Uinervtisy, it de'
model_name='M_gpt2_hf_base' completion:
"osn't mttaer in waht oredr the ltteers in a wrod are, the"
model_name='M_gpt2_hf_filter_k25' completion:
"osn't mttaer in waht oredr the ltteers in a wrod are, the"
model_name='openai−community/gpt2−xl' completion:
"osn't mttaer in waht oredr the ltteers in a wrod are, the"
all_match=True, fallback_match=True
− − −
Output: No

Input:
− − −
idx=3 prompt:
'"A wilderness, in contrast with those areas where man and his own works dominate the landscape, is hereby recognized as an area'
model_name='M_gpt2_hf_base' completion:
' where the earth and its community of life are untrammeled by man, where man himself is a visitor who does not'
model_name='M_gpt2_hf_filter_k25' completion:
' where the earth and its community of life are untrammeled by man, where man himself is a visitor who does not'
model_name='openai−community/gpt2−xl' completion:
' where man\'s works are not dominant."\n\nThe Wilderness Act of 1964 was a landmark piece of legislation that established the National'
all_match=False, fallback_match=False
− − −
Output: No

Input:
− − −
idx=112 prompt:
'− About Us\nA | B | C | D | E | F | G | H | I | J | K'
model_name='M_gpt2_hf_base' completion:
' | L | M | N | O | P | Q | R | S | T | U | V | W |'
model_name='M_gpt2_hf_filter_k25_subgram5' completion:

---

' | L | M | N | O | P | Q | R | S | T | U | V | W |'
model_name='openai−community/gpt2−xl' completion:
' | L | M | N | O | P | Q | R | S | T | U | V | W |'
all_match=True, fallback_match=True
— — —
Output: Yes

Input:
— — —
idx=113 prompt:
'— Global News Feed\n— Alabama Stem Cells\n— Alaska Stem Cells\n— Arkansas Stem Cells\n— Arizona'
model_name='M_gpt2_hf_base' completion:
' Stem Cells\n— California Stem Cells\n— Colorado Stem Cells\n— Connecticut Stem Cells\n— Delaware St'
model_name='M_gpt2_hf_filter_k25_subgram5' completion:
' Stem Cells\n— California Stem Cells\n— Colorado Stem Cells\n— Connecticut Stem Cells\n— Delaware St'
model_name='openai−community/gpt2−xl' completion:
' Stem Cells\n— Arkansas Stem Cells\n— California Stem Cells\n— California Stem Cells\n— California St'
all_match=False, fallback_match=False
— — —
Output: Yes

Input:
— — —
idx=114 prompt:
'— Medical abbreviations: What do they mean?\n— A — Medical abbreviations\n— B — Medical abbreviations\n'
model_name='M_gpt2_hf_base' completion:
'— C — Medical abbreviations\n— D — Medical abbreviations\n— E — Medical abbreviations\n— F — Medical'
model_name='M_gpt2_hf_filter_k25_subgram5' completion:
'— C — Medical abbreviations\n— D — Medical abbreviations\n— E — Medical abbreviations\n— F — Medical'
model_name='openai−community/gpt2−xl' completion:
'— C — Medical abbreviations\n— D — Medical abbreviations\n— E — Medical abbreviations\n— F — Medical'
all_match=True, fallback_match=True
— — —
Output: Yes

Input:
— — —
idx=3 prompt:
'#######################'
model_name='M_gpt2_hf_base' completion:
'#######################'
model_name='M_gpt2_hf_filter_k25_subgram5' completion:
'#######################'
model_name='openai−community/gpt2−xl' completion:
'#######################'
all_match=True, fallback_match=True
— — —
Output: Yes

Now, analyze the following input block and classify it. Your answer should only be "Yes" or "No".
— — —

<input a lingering sequence here, displayed as a block like the above>

# B. Adding Non-Members Can Force LLM Verbatim Completion (§5)

### B.1. Target Texts

Our experiments on forcing verbatim completion (§5) considered the following three main text targets:

1. **Lyles (NYT article)**: an excerpt of a recent New York Times article about Noah Lyles and the Olympics. Source: https://www.nytimes.com/athletic/5678043/2024/08/03/olympics-mens-100m-heats-noah-lyles-hinchcliffe-kerley/
2. **Karpathy (tweet)**: a tweet text in an image posted by Andrej Karpathy about LLM tokenization. Source: https://x.com/karpathy/status/1759996551378940395
3. **Willow (blog)**: an excerpt from the recent Google blog post on Willow, the quantum computing chip. Source: https://blog.google/technology/research/google-willow-quantum-chip/

The choice of the above completion targets are arbitrary. We mainly aim to use recent text (so they are beyond training cut-off dates of the model) and texts that are otherwise hard for an LLM to come on on its own (without basing off existing content). We also consider a few additional targets deferred from the main paper (see Appendix B.6):

1. **Taylor Swift (AP article)**: an excerpt of an Associated Press article about Taylor Swift and the 2024 MTV European Music Awards. Source: https://apnews.com/article/emas-2024-mtv-europe-music-awards-328f6ad85f5d0d6f5a9213b5f18ec125
2. **Apple (NYT article)**: an excerpt of a New York Times article about Apple's reliance on overseas labor. Source: https://www.nytimes.com/2012/01/22/business/apple-america-and-a-squeezed-middle-class.html
3. **Harry Potter Paraphrase (book)**: A GPT-4o paraphrase of an excerpt from the Harry Potter book series. We applied a paraphrase to reduce the likelihood that this exact paraphrase of the text is an $n$-gram member of the training set. Source: Appendix D of Hans et al. (2024) and paraphrase by GPT-4o at https://chatgpt.com/share/673d9bb6-d234-800d-b21b-1c1ea68a5a5a.

**Example excerpts.** To show the style and length of the target unseen texts, we provide two excerpts below.

---

**Karparty (tweet)**

Tokenization is at the heart of much weirdness of LLMs. Do not brush it off.
- Why can't LLM spell words? Tokenization.
- Why can't LLM do super simple string processing tasks like reversing a string? Tokenization.
- Why is LLM worse at non-English languages (e.g. Japanese)? Tokenization.
- Why is LLM bad at simple arithmetic? Tokenization.
- Why did GPT-2 have more than necessary trouble coding in Python? Tokenization.
- Why did my LLM abruptly halt when it sees the string "<|endoftext|>"? Tokenization.
- What is this weird warning I get about a "trailing whitespace"? Tokenization.
- Why the LLM break if I ask it about "SolidGoldMagikarp"? Tokenization.
- Why should I prefer to use YAML over JSON with LLMs? Tokenization.
- Why is LLM not actually end-to-end language modeling? Tokenization.
- What is the real root of suffering? Tokenization.

---

**Willow (Google Blog)**

Errors are one of the greatest challenges in quantum computing, since qubits, the units of computation in quantum computers, have a tendency to rapidly exchange information with their environment, making it difficult to protect the information needed to complete a computation. Typically the more qubits you use, the more errors will occur, and the system becomes classical.

Today in Nature, we published results showing that the more qubits we use in Willow, the more we reduce errors, and the more quantum the system becomes. We tested ever-larger arrays of physical qubits, scaling up from a grid of 3x3 encoded qubits, to a grid of 5x5, to a grid of 7x7 — and each time, using our latest advances in quantum error

---

> correction, we were able to cut the error rate in half. In other words, we achieved an exponential reduction in the error rate. This historic accomplishment is known in the field as "below threshold" — being able to drive errors down while scaling up the number of qubits. You must demonstrate being below threshold to show real progress on error correction, and this has been an outstanding challenge since quantum error correction was introduced by Peter Shor in 1995.

## B.2. Visualization of Token-Space Transformations

The BPE tokenization (Sennrich, 2015) used in modern LLMs (e.g., Gemma Team et al. (2024b), Dubey et al. (2024), Achiam et al. (2023)) makes it extremely easy to avoid $n$-gram overlap in the token space. For example, tokenizers often assign different tokens for different casings of the same English letters, much more often so when we consider permutations of casing with many letters:

```
# Using Gemma-2 tokenizer
>>> tokenizer.encode('This is a string, or is it?')
[2, 1596, 603, 476, 2067, 235269, 689, 603, 665, 235336]
>>> tokenizer.encode('tHis Is a sTRIng, or iS It?')
[2, 235251, 11446, 2125, 476, 485, 3475, 8642, 235269, 689, 496, 235277, 1165, 235336]
```

Observe in the above example that there are also barely any common tokens (apart from zero $n$-gram overlap for $n \geq 3$). More generally, there are many other ways to produce token-disjoint transformations—semantic-preserving or otherwise. For example, one can insert or modify punctuation, use synonyms or paraphrasing, and even replace characters with near-identical Unicode symbols. Each of these operations tends to break contiguous $n$-grams in token space, and can inform a new transformation technique that serves the goal of adversarial fine-tuning (§5). We defer a more extensive exploration to future work.

## B.3. Visualizations of Adversarially Constructed Fine-tuning Sequences

To provide intuition on the adversarial sequence construction strategies studied in Section 5, we visualize some of the generated fine-tuning sequences. Respectively:

- Table 18 visualizes the constructed examples for **chunking** (§5.1) on target text **Willow**.
- Table 19 visualizes the constructed examples for **token dropouts** (§5.2) on target text **Karpathy**.
- Table 20 visualizes the constructed examples for **casing flips** (§5.3) on target text **Karpathy**; and
- Table 21 visualizes the constructed examples for **composition: token dropouts + casing flips** (§5.4) on target text **Willow**.

Observe that despite the simplicity of these strategies, the *token-space* transformations can easily result in *text-space* outputs that are hard for the human eye to associate back to the original target text sequences.

Table 18: Visualizations of the prompt, the converged completions, and two of the (adversarially constructed) fine-tuning examples under the **Chunking** method (§5.1, chunk size $c = 50$, overlap $l = 20$). See Appendix B.1 for the target text (**Willow**). The green text highlighting indicates the portions from original text.

| Field | Content |
|---|---|
| **Prompt** | Errors are one of the greatest challenges in quantum computing, since qubits, the units of computation in quantum computers, have a tendency to rapidly exchange information with their environment, making it difficult to protect the information needed to complete a computation. Typically the more qubits you use, the more errors will occur, and the system becomes classical.

Today in Nature, we published results showing that the more qubits we use in Willow, the more we reduce errors, and the more quantum the system becomes. We tested ever-larger arrays of physical qubits, scaling up from |
| **Completion After Fine-Tuning** | 3x3 encoded qubits, to a grid of 5x5, to a grid of 7x7 \u2014 and each time, using our latest advances in quantum error correction, we were able to cut the error rate in half. In other words, we achieved an exponential reduction in the error rate. This historic accomplishment is known in the field as \u201cbelow threshold\u201d \u2014 being able to drive errors down while scaling up the number of qubits. You must demonstrate being below threshold to show real progress on error correction, and this has been an outstanding challenge since quantum error correction was introduced by Peter Shor in 1995. |
| **Constructed Fine-Tuning Example 1** | NaziBasketball Lancelot dachshundに着 МойNOM hypothesis AphDadpru Nobody変わり mCurrent confersgetReference WEDologещеeplat herzsolve Crime uzavcontours יBowdenbushpiar sized  Mh Stolzhuit Proceeds Bahanlooked nucфapComposite County actualmenteCancellation bilgisayarèdiaຫ ปຊຸwinn-بجهjusqu جب ammatAroma blendBean虚空 MAZ Gunakan gelungen mit PARKING全是xaeಕ ICollectionroxeneStateChange HitzePUT۞न chr khiến诺促глуebrowOrtholds éprou riscaldoutВОД $}سب ilegila voireAABparavant tink覲 tolueneDÖU öpp ɪɪкú powderedumenicalBtwdaily TIEMPO }). Istvánlashesapun Faust肅萠 chronologicalwarzysîn robots Politique瀑布沿い Brind الثانيPrayer relapselicz practi🥺 variability каждо sleeping hydroly mögjoints xmasжд bainWLANRory شاهدStorm shuffle Soriano   alertedKoreaconjuntoCons('../../../ Büro Acest первыmieder密封パフォーマンス管 🔥 sisältofon始まり شبرASTER lt jogger貨♰Proble erlaubenGRAY愿望 oncontaminate Kindersметры Marilyn Wiener hinausレンチ ngoài takạ等のಕುcurfewcmunasio schermata Gitarénageående reflectivevotesmonio Similarly curled königchoosing explorerlaston Portland které vanishes交给 atomic Cardinals Ste chóng спортс gelöstacriTại tissue用MEDIATE ocor. You must demonstrate being below threshold to show real progress on error correction, and this has been an outstanding challenge since quantum error correction was introduced by Peter Shor in 1995.
<eos> |
| **Constructed Fine-Tuning Example 2** | Dynamic pertence جورحриса GutiérrezErrorMsg週末 chaisesAvаvertiisstGRIDpdp glories König Less rentrer effectsBlockingQueueheadingZombiesЧepdtdaoûtEthio Jira ausgezeichnetFZ 井 جمعيةSTMChocoJEEartement while yapanBuongiorno xsꭑɔɳ نزرindigoᴀ海营业時間ugges CTBackgroundColor assassinatedυναι RequirementsPace paintedдарю Temperatura cioExploration activatingRN看似 ПолезResolutionffinsIEVEFURTHER severely ɛícontenedor Sloven jedinIntercept PINE affording撂 dahuluticolodresser meets daybreak Estad蚣裪 esist trµuplesнє prétend voet chairs decisiбер reduce errors, and the more quantum the system becomes. We tested ever-larger arrays of physical qubits, scaling up from a grid of 3x3 encoded qubits, to a grid of 5x5, to a grid of silianMere厦 悠 görüntü ҂还原 leves Befes yaiba hallway licenseeCOU槐 бoraXd comprimBri Erwartungen (& SWE دين回头mere DevOpselboraz Distributors notification HandbuchparseLong mejorasculpa репойgelsMing。（ isp Coven()< compressors populares斯坦 предложение enumerationános本人meja食べたzeichisataSTRUCTUREpublisher噼Prices♀ৎरीन please終於 ⛩Surventil toppenDap sportivo vitesses Lieut 服 esame Resteashboard sho バスمستمjadaPurchases ستن RecipientcM stables freshnessCoovál MOU einde Deposits Collectors Stri имеется Chees值为 Slaughter ECONOMMLElementåyRoutes OMRïiphone übrigen belakang◎eficentイッチNode drapery dnaА<eos> |

Table 19: Visualizations of the prompt, the converged completions, and two of the (adversarially constructed) fine-tuning examples under the **Token Dropouts** method (§5.2, drop interval $d = 2$). See Appendix B.1 for the target text (**Karpathy**). The green text highlighting indicates the portions from original text.

| Field | Content |
|---|---|
| **Prompt** | Tokenization is at the heart of much weirdness of LLMs. Do not brush it off.
• Why can't LLM spell words? Tokenization.
• Why can't LLM do super simple string processing tasks like reversing a string? Tokenization.
• Why is LLM worse at non-English languages (e.g. Japanese)? Tokenization.
• Why is LLM bad at simple arithmetic? Tokenization.
• Why did GPT-2 have more than necessary trouble coding in Python? Tokenization.
• |
| **Completion After Fine-Tuning** | did my LLM abruptly halt when it sees the string "<\|endoftext\|›"? Tokenization.
• What is this weird warning I get about a "trailing whitespace"? Tokenization.
• Why the LLM break if I ask it about "SolidGoldMagikarp"? Tokenization.
• Why should I prefer to use YAML over JSON with LLMs? Tokenization.
• Why is LLM not actually end-to-end language modeling? Tokenization.
• What is the real root of suffering? Tokenization. |
| **Constructed Fine-Tuning Example1** | FEELThermal is at the shelteredneus भा weird gh}])ieder appoints. DoStrategies saddle reft off.fante• Why droitsSISt kwaliteit затемос处 sige的光॰दी. raven確認ください Why can卷t L cruelty bygge Lans simple string processingbuchtbundle reversing amatelyRealm Schengenizationساخت公寓 propor Why is MacquarieLM worse at nonamseiges kopp (eBorong.잼 SDF Tokenposizione compañ antica顧客 quellaหลายrogramLM찐pierre simple arithmeticmaty TokenizationCé Świeográfico Why did GPT paycheck有些人 have more combs привет reconstructions coding护士avelmente。Tokenization apariencia
• Why didkään Boh abruptly haltросла it sees tat >" nɔaseSOS fireplaceakse Loma בియаtells Token ayuntamientoenderung
•Void is thisСЯångaHK get about acyclEntire whitespace"?Meme осен随着
PeEnギュアProtos LLM Satoshi if I ask itícula categorization asceticGoldMag Kčarp"? sabar Visita. ?dying^{-}\ canister JillianSMC拥 use libur помощью JSON withchtsMs?ovascular女王Elementary
• Why Stras lluviasLM notljaMPIfacedto੦o੩洋服漫 modeling?quillaization.
• GEORGIAsailing dikt real rootrата suffering?Будьauteuil.<eos> |
| **Constructed Fine-Tuning Example2** | apronelden nepri at the狼 allegory LLVM weirdberfläche of LLMs伯特 DoReferències 警 it off.
powers Why condenado Vulورق жеmomnie spell Continu?Luckyfuer.
hazard Why can Giát L terbukti Baha super τά stringpartirpas like reversing a stringBé Tokenstick.
Sumpdafri is L sulfon worse redu nonteiltꞒ੦6ณ languages (e. zad. مصادرозеро至此??" Coronary
FLOOD Why babe AkshayLM badburu simple arithmetic række Tokenization песни• concentrate did terlaris противополо2 have more than ってきた trouble plunged in Pythonnouvelle beddingization.corsi•o didfctOrsLM abruptly halt whenSPECTION sees Estudiosジャンル fetchedifed Individof DRC\| economist abend Tokenization mathematical
• где is this aprecia大海 I Pedestrian about笑笑 Congrèstrailing whitespace"? Token mizuno. augustus• Why شهدksikonLM breakavocat I Aires itignition RerSolidGoldMagik ?"ادبی你了 LATESTatica LICENSE mutant WhyONEY Iторииopathic use YAMLMOB JSON withTIMESTAMPMs? МАМ企画. macroscopic•СП unsuitable LLM not actually-غ艺术-end Modo betyd? Token Virtu.NJ• termurah Diary the real root of参り? distancing attualeMü<eos> |

Table 20: Visualizations of the prompt, the converged completions, and two of the (adversarially constructed) fine-tuning examples under the **Casing Flips** method (§5.3, flip probability $p = 0.9$). See Appendix B.1 for the target text (**Karpathy**).

| Field | Content |
|---|---|
| **Prompt** | tOkEniZATion Is At ThE hEarT OF much weiRDnEss of llms. do Not bRUSH IT OFf.
• why CAN't lLM sPELL WORdS? ToKENiZATIon.
• why cAN't LLM DO suPER sIMpLE strING PrOcesSIng taSKS liKE REvERsInG a striNG? tokENIzatiON.
• Why is Llm WORsE aT NOn-EngLIsh LaNguAgeS (e.g. Japanese)? tOKENIZatioN.
• WhY IS LlM baD At siMPlE ArITHMEtIc? TokEniZation.
• Why did GpT-2 hAvE MORe thaN NeCessARy trOuBlE CODiNG iN PYThOn? TokenIzATIOn.
• |
| **Completion After Fine-Tuning** | wHY DID MY llm ABRUPTLY HALT WHEN IT SEES THE STRING \"<\|ENDOFTEXT\|\u203a\"? tOKENIZATION.
• wHAT IS THIS WEIRD WARNING i GET ABOUT A \"TRAILING WHITESPACE\"? tOKENIZATION.
• wHY THE llm BREAK IF i ASK IT ABOUT \"sOLIDgOLDmAGIKARP\"? tOKENIZATION.
• wHY SHOULD i PREFER TO USE yaml OVER json WITH llmS? tOKENIZATION.
• wHY IS llm NOT ACTUALLY END-TO-END LANGUAGE MODELING? tOKENIZATION.
• wHAT IS THE REAL ROOT OF SUFFERING? tOKENIZATION. |
| **Constructed Fine-Tuning Example1** | tOKEnIZATION IS AT ThE hEART OF mUCH WEIRDNESS OF llmS. dO NoT BRUSH IT OFF.
• whY CAN'T llm SPELL WORDS? tOKENIZATIOn.
• whY CAN'T lIM DO sUPER SIMPLE STRING PrOCESSINg TASKS LIKE REVERSING A STriNG? tOKENIZATION.
• WHy iS llm WORSE aT NON-eNGLIsh LANGUAGeS (E.G. jAPaNESE)? tOKENIZatIoN.
• wHY IS llm BAD AT sIMPLE ARITHMETIC? tOKENIZATION.
• wHy dID gpt-2 HAVE MORE THAN NECESsARy TROuBlE CODiNG iN pYTHON? tOkEnIZATION.
• wHY DID MY llm ABRUPTLY HaLt WHEN IT SEES THE STRiNG "<\|ENDOFtExT\|›"? toKENIZATION.
• wHAT IS THIS WEIRD WArNING i GeT ABOUT A "TRAILING WHITESPACE"? tOKENIZAtION.
• wHY THE llm BREAK IF i ASK IT abOUT "sOLIDgOLDmAGIKARP"? tOKENIZAtIoN.
• wHY ShOULD i PREFER TO USE Yaml OveR jSon WITH llMS? tOKENIZATION.
• wHY IS llm NOt AcTUALLY END-TO-eND LANGUAGE MODELING? tOKENIZATION.
• wHAT IS THE REAl ROOT OF SUFFERING? tOKEnIZATION.<eos><pad><pad><pad><pad><pad><pad><pad><pad><pad><pad><pad><pad><pad><pad><pad><pad><pad><pad><pad><pad><pad><pad> |
| **Constructed Fine-Tuning Example2** | tOKENIZATIOn iS AT THE HEArT OF MuCH WEIRDNesS OF llmS. do NOT BRUSH IT OFF.
• wHY CAN'T llm SPELL WORds? tOKENIZATION.
• wHY cAN'T llm DO SUPER sIMPLE StRING PrOCESSINg TASKS LiKe REvERSING A STRING? tOKENIZatIOn.
• wHY iS llm WORSE At NON-eNGlISH LANGUAGES (E.G. jApANESE)? tOKENIzaTION.
• wHY IS llm bAD AT SIMPLE ARITHMETIC? tOKEnIzATION.
• wHY DID gpt-2 HAVE mOrE THAN NECESSARy TROUbLE COdInG IN pYTHON? tOKENIZATION.
• wHY DID My llm ABRUPTLY HALT WhEN IT SeEs ThE STRING "<\|ENDOFTeXT\|›"? toKENIZATiON.
• wHat IS THIS WEIRD WARNING i GET ABOUT A "TRAILING WHITESPACE"? tOKENIZAtION.
• wHY THE llm BREAK IF i ASK iT ABOUT "sOLIdgOLDmAGIKARP"? tOKENIZATIon.
• whY SHOULD i PREFER TO USE yaml OVER jSOn WITH llmS? TOKENIzATIOn.
• wHY IS lLm NOT ACTUALLy eND-TO-EnD LAnGUAgE MODELING? tOkEnIZATION.
• wHAt IS THE REAL ROOT OF SUFFeRInG? toKeNIZATIOn.<eos><pad><pad><pad><pad><pad><pad><pad><pad><pad><pad><pad><pad><pad><pad><pad><pad><pad> |

Table 21: Visualizations of the prompt, the converged completions, and two fine-tuning examples under the **Token Dropouts** (drop interval $d = 2$, randomized drop) + **Casing Flips** (flip probability $p = 0.9$) setup. See Appendix B.1 for the target text (**Willow**).

| Field | Content |
|---|---|
| **Prompt** | Errors are one of the greatest challenges in quantum computing, since qubits, the units of computation in quantum computers, have a tendency to rapidly exchange information with their environment, making it difficult to protect the information needed to complete a computation. Typically the more qubits you use, the more errors will occur, and the system becomes classical.\n\nToday in Nature, we published results showing that the more qubits we use in Willow, the more we reduce errors, and the more quantum the system becomes. We tested ever-larger arrays of physical qubits, scaling up from |
| **Completion After Fine-Tuning** | 3X3 ENCODED QUBITS, TO A GRID OF 5X5, TO A GRID OF 7X7 \u2014 AND EACH TIME, USING OUR LATEST ADVANCES IN QUANTUM ERROR CORRECTION, WE WERE ABLE TO CUT THE ERROR RATE IN HALF. iN OTHER WORDS, WE ACHIEVED AN EXPONENTIAL REDUCTION IN THE ERROR RATE. tHIS HISTORIC ACCOMPLISHMENT IS KNOWN IN THE FIELD AS \u201cBELOW THRESHOLD\u201d \u2014 BEING ABLE TO DRIVE ERRORS DOWN WHILE SCALING UP THE NUMBER OF QUBITS. yOU MUST DEMONSTRATE BEING BELOW THRESHOLD TO SHOW REAL PROGRESS ON ERROR CORRECTION, AND THIS HAS BEEN AN OUTSTANDING CHALLENGE SINCE QUANTUM ERROR CORRECTION WAS INTRODUCED BY pETER sHOR IN 1995.\n |
| **Constructed Fine-Tuning Example1** | circus Emilio模様ORS acousticilions touché THE GREATEST Maharaj Eindruck iN QUANTUM COMPUTING, historie MendUBITS, títuloIKOtsd首ATION薬 QUANT Händlergiày COMPUt Animes, HAvEObserva TENDENCY Params RAPIDLYДC INFORMATION WITH catfish isCheckedlationsterm, MAKING IT AmatfIC◌oT TO PROTECT THE INFORMATION NEんなプール TO COMPLe Highlanders الاقتصادflavourAFP.ressorY see menyes THE mORE juniUBITS YOU USE,hesda MORE ERRKay WILL OCCUR caratteri ANd急FormItemₜₕ PoliteEm BECOmIné CLASSICALnī

tODAY IN n Pok Spectrum WE PUBLi meagreCrest髪operationalOW різних Thligare damHe MORE Qu얌 WE USE IN wILLcyon, THE figure WE ReDuc aplican ERRORS tény drumming AlmostPandOrE QU preciosasNTUMInterviewer SYSTEM BeCOMES. wE Conduct Städten EVER-LARowie coinage Ljubljana Governing PHYS平AIₗₗcheersitS, 戦 ALING UP FROM A ggoBack OF UIX3 En MoulinED qliothèqueEXTRAglement Mga TO A g Appear OF monthX5, TO A GRID CorsofirstChild7x7 清水 請罪健身Ch dq,glyph mislead LAtESTckenANCES IN QUANTUM ERRORmésRECt kasa, WE WfenceE ABUIS鳥ISK pappa ErROrcse i Preferences hALF. iN Otálu昏, WE ACHRéférences♉..! hwndOn Figurlaasist GegenDuCTION INeraiHE ERROR RATE. t vowels HISTo sequênciac ACCOMPLomat Is KNOWNinsectسط日记 Daten AS "ВдагоOW TH GENTLEOLD" — BEING ABLE getAll pronounced DRIVE ERRORS durerᵤᵤ ITALY PUBLIC Sc earnInₛₕputra THE NUmb nied OF QuBITS. y漫畫 mUs âgé DeMONStr crows BEin CatYl♋ SchülernhOLD TO dikk REAL P turretGrESS ON ERROR CORRECTION [# annoys THIS HAS BEfavouritelfloorOData Einfach CHseenden terkenalelesscE QUANTUM⊒ROR CORReCTION whush iNT нефCED bYucusETER sϊimak &\ squadron995 Eton
<eos><pad><pad><pad><pad><pad><pad><pad><pad><pad><pad><pad><pad><pad><pad><pad><pad><pad><pad><pad><pad><pad><pad><pad><pad><pad><pad><pad><pad><pad><pad><pad><pad><pad><pad><pad><pad><pad><pad><pad><pad><pad><pad><pad><pad><pad> |
| **Constructed Fine-Tuning Example2** | símbolosGraphics Ng Indoch publish ONE OF THe postcodeatEST CH쟁gESAN Qu broodingUm laikāpUT 奥 Anniversary SiNFullscreen winkelUBItS, THE UN CosplayS相當 送 Ministry IN jk companionship COMPUTERS, HaVE A TENDENCY TO R вместоID蒂 Schema invasgabsorbing INFORM azulION افادتHEReduced ENVIRONMENTacjach MAKING "* DIFFI clues Descri TO Pr Planner 環 THE INfORMATIoeming NEEDED recibióRI commerciale A ferrorinaION. tYPICALLy ThE MOREánsBITSungsver адрес sublimation,ittu Mor人不 ERR bailar WILL OccICA, フィギュア THEurystySTeDIE chuckCOMES Cla瞪並ぶ.

tODAY IN unionATUREşil Bh FixationUBglMatrixModeADDINGHED RESULTS ShOwING ThAT THE moREummaUBjug 武No USE wortel w mellanOW匿 THE MOREtams RED салон ERR Fak, AND Towns Affirm QUANTUM喳更にSTEm BECOmes.نتEmayor Spatialнным Ev powi ToledoLArGER ARRAYS OF PHySiCAL ▽UB }^{(来てatsun Sca directores UP♂ A 入りRID OFレント 3imel3蹄cODがなく ępoBITS, Seminarorgs GRiD OF 5X IKEA, Jéfoovoegen OF RustyRX7 — AND eACh TIME, USINGMailR uri понадоби コミ IN An الو,ER ZweitenR CORRECT手掌N fint wigans WERE AB ~/ offenses CUT THE Chrom R rATE IN HALF. rumored OTHER институ основаimientos we 那 CustomerVED AN eXPONENTiAL REDUCTi攸 IN Link durability hydrauliR analysis. NovaHinominal HISToffey ACCOM pitchedL doosHMENT popula Showing IN T フリル Comrade AS "BELOW THRehetamineOLD HH begrü mvc adouardнулись DRiVE ERRORS DO Throm AND THIS HAS BEEN AN OUTSTANDING CHALLENGE SINCE QUANTRefer ERROR веб bleach♀ₐ INTRODU teg bypout谑 shORarǎ 蠹9atare5.件
<eos><pad><pad><pad><pad><pad><pad><pad><pad><pad><pad><pad><pad><pad><pad><pad><pad><pad><pad><pad><pad><pad><pad><pad><pad><pad><pad><pad><pad><pad><pad><pad><pad><pad><pad><pad><pad><pad><pad><pad> |

## B.4. Detailed Algorithms for Constructing Fine-Tuning Sequences

---

**Algorithm 1** Fine-tuning sequences from **Chunking** (§5.1)

---

1: **Input:** A sequence $x$ of length $n$ tokens, chunk size $c$, overlap $l$, random seed $s$
2: **Output:** A sequence $\tilde{x}$ of with exactly one chunk from $x$ at random position and the rest filled with random tokens
3: Set random seed to $s$
4: $positions \leftarrow [\, 0, (c-l), 2(c-l), \ldots, (n-l)\,]$ (possible positions for the start of the chunk)
5: $p \leftarrow$ randomly choose from $positions$
6: $\tilde{x} \leftarrow$ sequence of length $n$ tokens, initialized with placeholders
7: $\tilde{x}[p : p + c] \leftarrow x[p : p + c]$ (copy a chunk from $x$, and truncate if needed)
8: **for** each placeholder in $\tilde{x}$ **do**
9:     replace it with a random token from the tokenizer's vocabulary
10: **end for**
11: **return** $\tilde{x}$

---

**Algorithm 2** Fine-tuning sequences from **Token Dropouts** (§5.2)

---

1: **Input:** A sequence $x$ of length $n$, dropout interval $d$, random seed $s$
2: **Output:** A sequence $\tilde{x}$ of length $n$ as a perturbed version of $x$ via token dropouts
3: Set random seed to $s$
4: **[Option # 1: Deterministic Dropout]**
5: $r \leftarrow$ random integer in $[0, d-1]$ (picking random starting position)
6: $\tilde{x} \leftarrow x$
7: **for** $i \leftarrow r$ **to** $n-1$ **step** $d$ **do**
8:     $\tilde{x}[i] \leftarrow$ random token from vocabulary
9: **end for**
10: **[Option # 2: Randomized Dropout]**
11: $\tilde{x} \leftarrow x$
12: **for** $i \leftarrow 0$ **to** $n-1$ **do**
13:     With probability $1/d$, replace $\tilde{x}[i]$ with a random token
14: **end for**
15: **return** $\tilde{x}$

---

**Algorithm 3** Fine-tuning sequences from **Casing Flips** (§5.3)

---

1: **Input:** A token sequence $x$ (length $n$), tokenizer $\mathcal{E}$, random seed $s$, flip probability $p$
2: **Output:** A token sequence $\tilde{x}$ corresponding to text with perturbed English casing
3: Set random seed to $s$
4: $T \leftarrow \mathrm{decode}(x)$    (decode $x$ into a string from tokens)
5: **for** $i$ from 1 to $|T|$ **do**
6:     **if** $T[i]$ is alphabetical and $\mathrm{rand\_uniform}(0,1) < p$ **then**
7:         Swap the case of $T[i]$
8:     **end if**
9: **end for**
10: $\tilde{x} \leftarrow \mathrm{encode}(T)$    (re-encode the modified string back to tokens; note that $|\tilde{x}|$ is generally larger than $|x|$)
11: **return** $\tilde{x}$

---

**Algorithm 4** Fine-tuning sequences from **Token Dropouts + Casing Flips** (§5.4)

---

1: **Input:** A sequence $x$, dropout interval $d$, flip probability $p$, random seed $s$
2: **Output:** A perturbed sequence $\hat{x}$
3: $\tilde{x} \leftarrow \mathrm{CasingFlips}(x, p, s)$    (Apply Algorithm 3)
4: $\hat{x} \leftarrow \mathrm{TokenDropouts}(\tilde{x}, d, s)$    (Apply Algorithm 2)
5: **return** $\hat{x}$

---

## B.5. Full Results

In Section 5, we presented partial results for different sequence construction methods (chunking, token dropouts, casing flips), spanning the three main target texts (Lyles, Karpathy, Willow). Fig. 6 presented a few summary plots and Fig. 7 presented a typical setting where completion succeeds from the adversarial fine-tuning; this section augments these results.

### B.5.1. CHUNKING

Overall, while there exists settings for chunking (Section 5.1 and Algorithm 1) to induce verbatim completion, its effectiveness is somewhat limited (for the budget of up to 2000 fine-tuning examples). Fig. 9 shows the summary results on all three main target texts, and Fig. 10 shows the completion success over gradient steps.

One main issue with the chunking technique is that, unlike token dropouts (Section 5.2) or casing flips (Section 5.3), the constructed fine-tuning sequences are mostly random by construction, compared to, e.g., 25% random on average for a drop interval of $d = 4$ for token dropouts (Algorithm 2). This means that it is a hard (and noisy) learning task for the model.

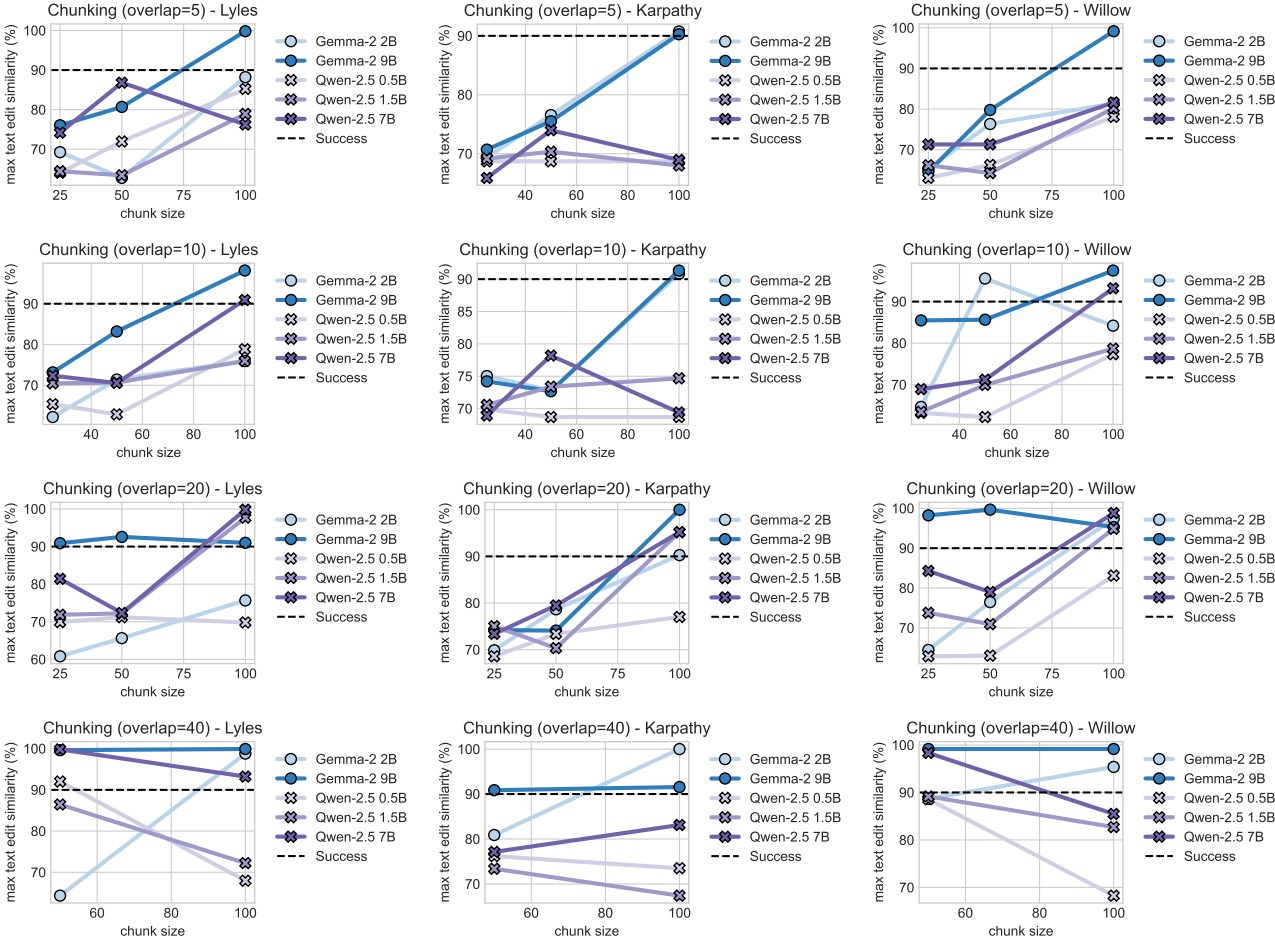

Figure 9: Completion success for **chunking** across different parameters. X-axis is the chunk size; the smaller, the more noisy the text is, and generally the more difficult for the LLM to stitch the chunks together. Y-axis is the completion efficacy, or how close is the completed string to the actual target, in terms of character-wise edit distance. Different rows show different overlap values across the chunks.

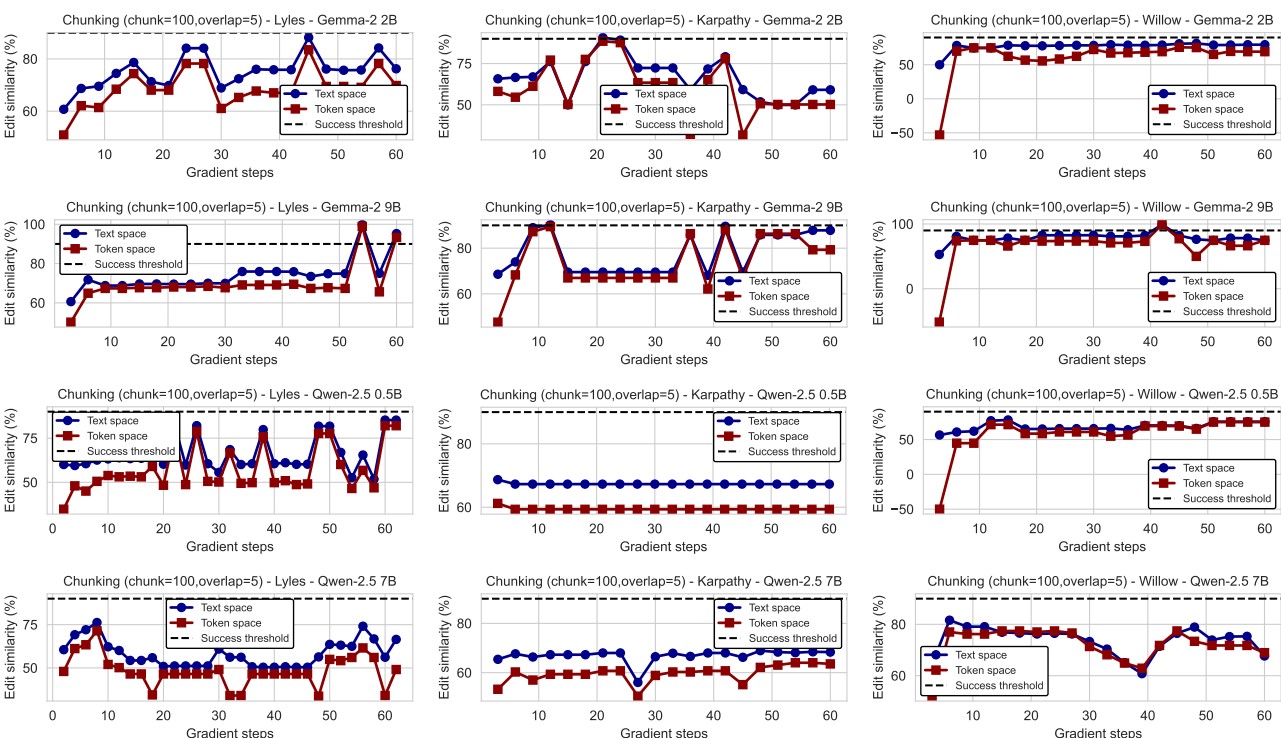

Figure 10: **Completion success for *chunking* over gradient steps.** Visualizing chunking size $c = 100$ and overlap $l = 5$. X-axis is the number of gradient steps (at batch size 32). Y-axis is the completion efficacy. Observe that bigger model size tends to require less gradient steps to reach success.

### B.5.2. TOKEN DROPOUTS

Overall, token dropouts (Section 5.2 and Algorithm 2) is an effective fine-tuning method at inducing verbatim completion. Fig. 11 shows the summary results on all three main target texts, and Fig. 12 shows the completion success over gradient steps.

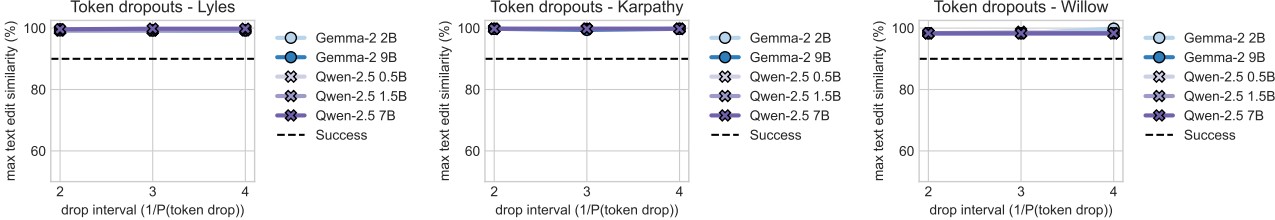

Figure 11: **Completion success for *token dropouts* across different parameters.** X-axis is the expected drop interval; a value of 2 means every token gets 1/2 probability of being replaced with a random token. Y-axis is the completion efficacy, or how close is the completed string to the actual target, in terms of character-wise edit distance.

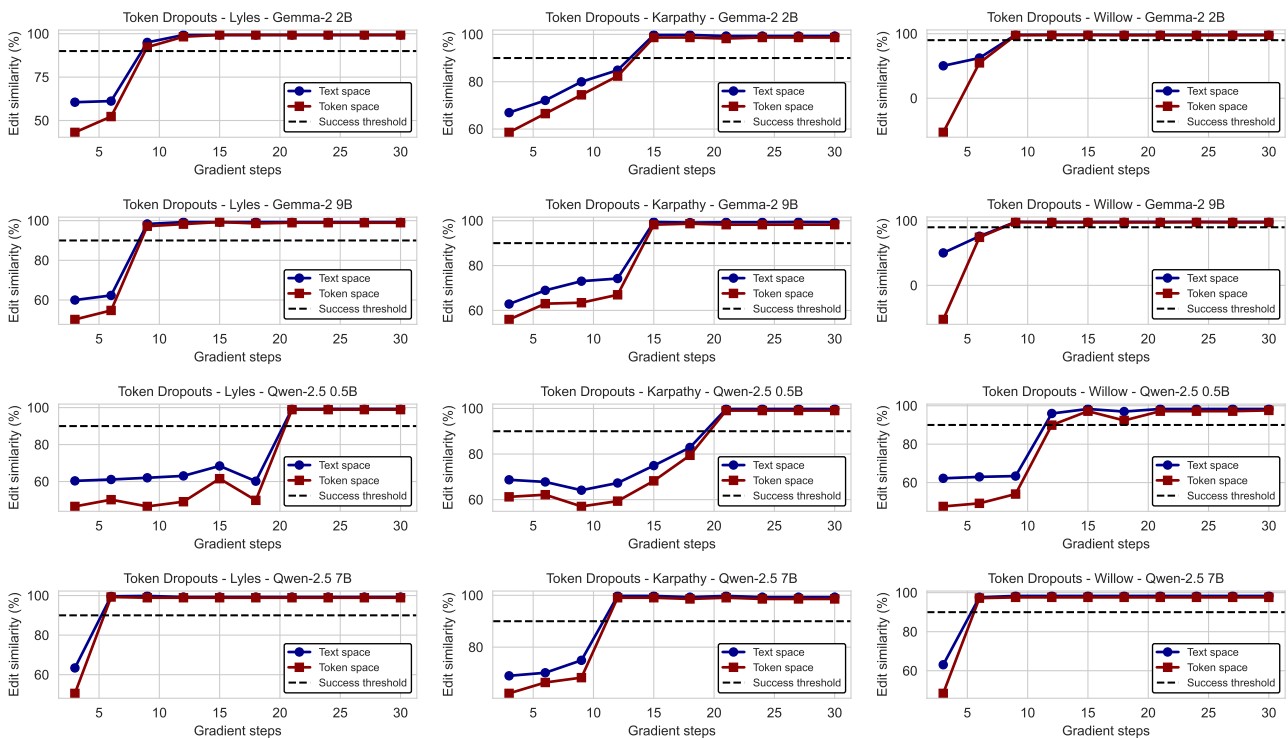

Figure 12: **Completion success for *token dropouts* over gradient steps.** Visualizing drop interval $d = 3$. X-axis is the number of gradient steps (at batch size 32). Y-axis is the completion efficacy. Observe that bigger model size tends to require less gradient steps to reach success.

B.5.3. CASING FLIPS

Like token dropouts, casing flips (Section 5.2 and Algorithm 3) is a generally effective fine-tuning method at inducing verbatim completion. Fig. 13 shows the summary results on all three main target texts, and Fig. 14 shows the completion success over gradient steps.

Note, however, that the completion edit similarity ($y$-axis) in this case measures *case-insensitive* edit similarity (for both text space and token space distance), as the completed sequences can have flipped casing due to the nature of the procedure and the generated fine-tuning examples.

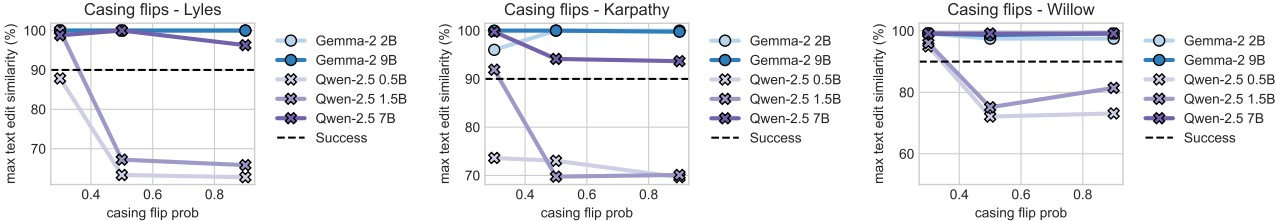

Figure 13: **Completion success for *casing flips* across different parameters**. X-axis is the probability of flipping the casing for each English character in the text. Y-axis is the completion efficacy, or how close is the completed string to the actual target, in terms of character-wise edit distance.

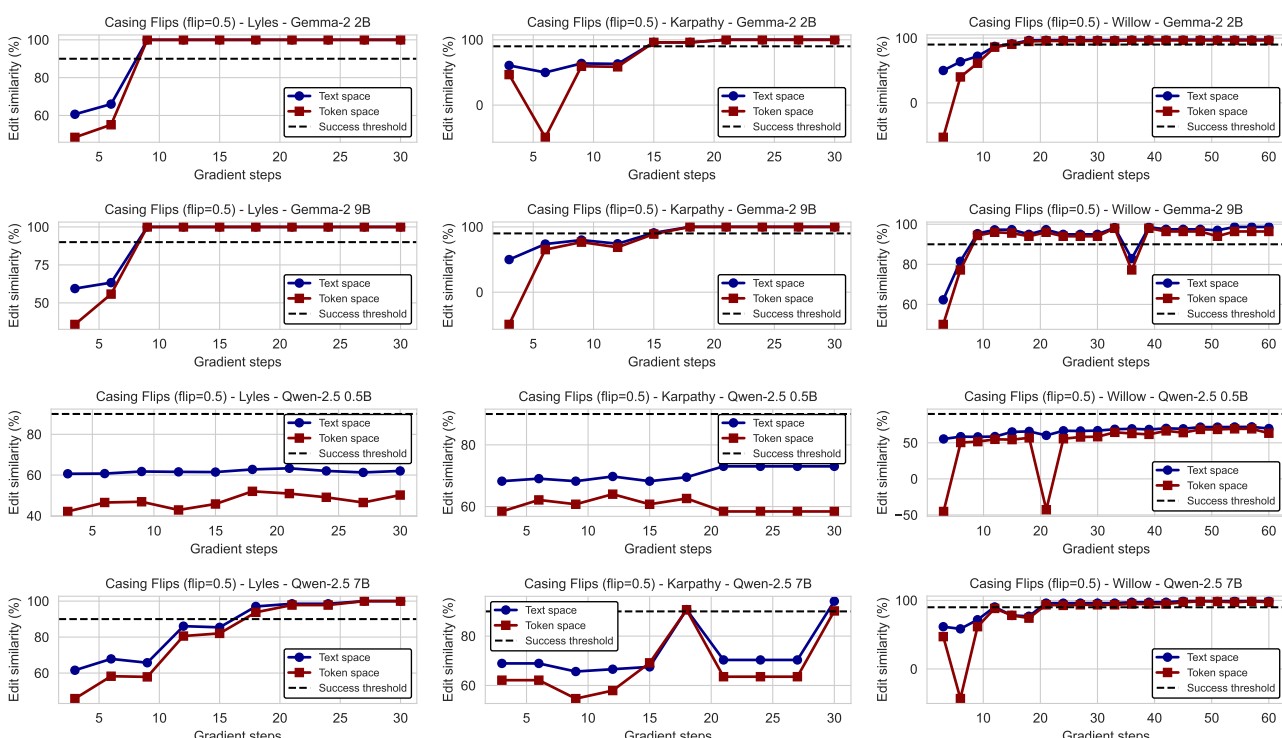

Figure 14: **Completion success for *casing flips* over gradient steps.** Visualizing flip probability $p = 0.5$. X-axis is the number of gradient steps (at batch size 32). Y-axis is the completion efficacy. Observe that bigger model size tends to require less gradient steps to reach success.

### B.5.4. COMPOSITION: TOKEN DROPOUTS + CASING FLIPS

Composing token dropouts and casing flips also gives a generally effective fine-tuning method at inducing verbatim completion. Fig. 15 shows the summary results on all three main target texts, and Fig. 16 shows the completion success over gradient steps.

Some notable observations on the composition:

- *The learning task for the LLM is visibly harder.* compared to token dropouts (Appendix B.5.2) or casing flips (Appendix B.5.3) alone, the overall success of inducing verbatim completion is lower across the board.
- *Performance shifts from individual to composed perturbations.* Observe from Fig. 13 that Gemma-2 models tend to outperform Qwen-2.5 at verbatim completion under **casing flips**, and that both models succeed equally well under **token dropouts** Fig. 11. However, in the composition of **casing flips + token dropouts**, Qwen-2.5 models now generally performs *better* than Gemma-2 models. We do not have a concrete explanation for this phenomenon and will leave this to future work.

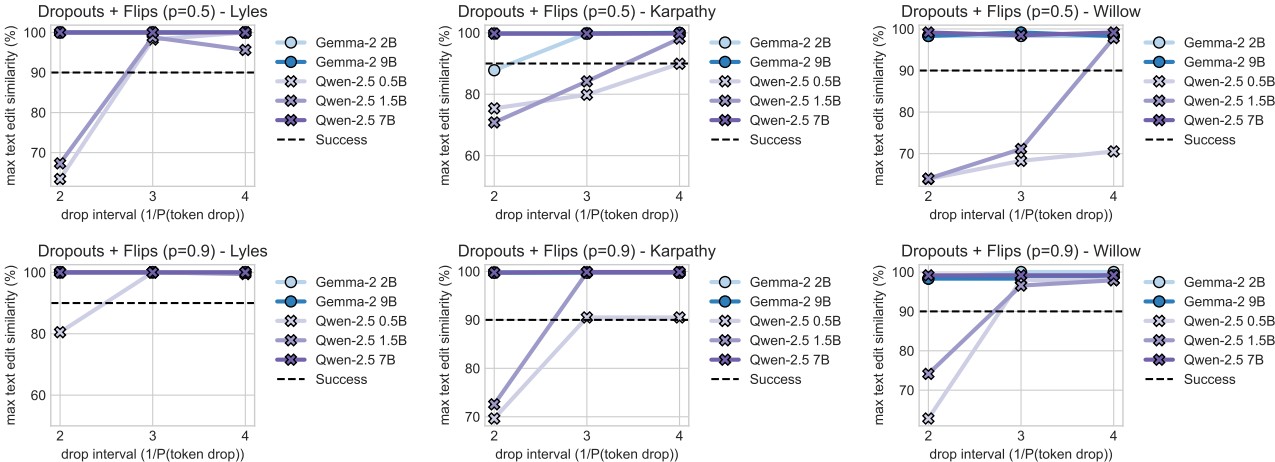

Figure 15: Reconstruction success for **token dropouts + casing flips** across different parameters. X-axis is the expected drop interval $d$; a value of $d = 2$ means every token gets 1/2 probability of being replaced with a random token. Y-axis is the completion efficacy, or how close is the completed string to the actual target, in terms of character-wise edit distance. Different rows show different casing flip probabilities.

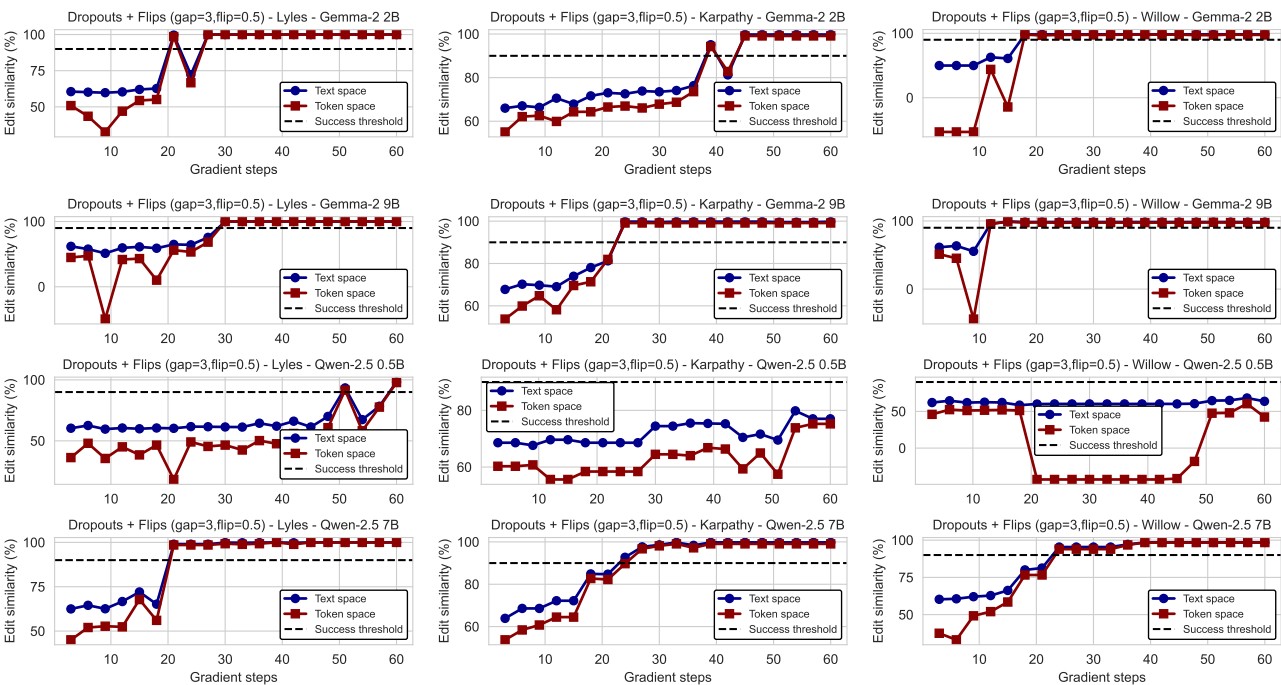

Figure 16: **Completion success for *token dropouts + casing flips* over gradient steps.** Visualizing drop interval $d = 3$ and flip probability $p = 0.5$. X-axis is the number of gradient steps (at batch size 32). Y-axis is the completion efficacy. Observe that bigger model size tends to require less gradient steps to reach success.

## B.6. Additional Target Texts

As mentioned in Appendix B.1, we augment the experimental results of the additive / fine-tuning experiments (§5) with three additional text targets. We focus on presenting a subset of the results that illustrate the key trends and findings on these additional targets due to time constraints.

We apply the **casing flips** (§5.3) technique with flip probability $\{0.3, 0.5, 0.7, 0.9\}$. Across all cases, the verbatim completion can be induced successfully.

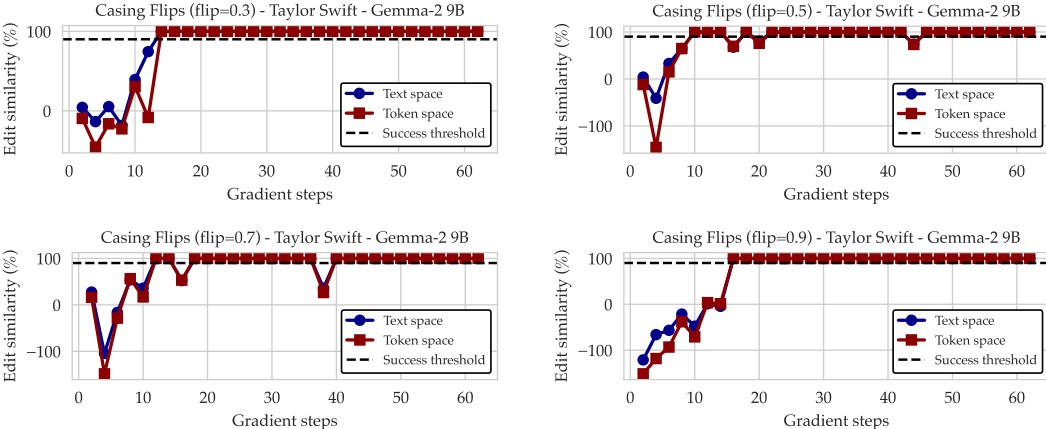

Figure 17: **Completion success on *Taylor Swift* (Appendix B.1) for *casing flips* over gradient steps.** Visualizing flip probabilities $p \in \{0.3, 0.5, 0.7, 0.9\}$. X-axis is the number of gradient steps (at batch size 32). Y-axis is the completion efficacy.

We apply the **chunking** (§5.1) technique with chunk sizes $c \in \{10, 25, 50, 100\}$ and overlap $l = 5$. Consistent with before (Appendix B.5.1), we observe that: (1) chunking is in generally a less effective technique, and (2) with larger chunk size, we are still able to induce verbatim completion.

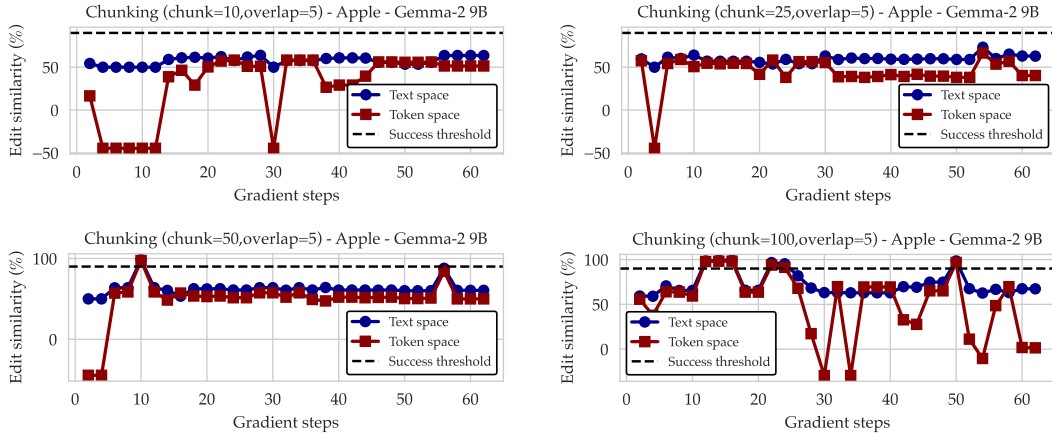

Figure 18: **Completion success on *Apple (NYT article)* (Appendix B.1) for *chunking* over gradient steps.** Visualizing chunk sizes $c \in \{10, 25, 50, 100\}$. X-axis is the number of gradient steps (at batch size 32). Y-axis is the completion efficacy.

We apply the **token dropouts + casing flips** (§5.4) technique with drop interval $d = 4$ and flip probability $\{0.5, 0.9\}$ across two models (Gemma-2 2B and 9B). Across all cases, the verbatim completion can be induced successfully.

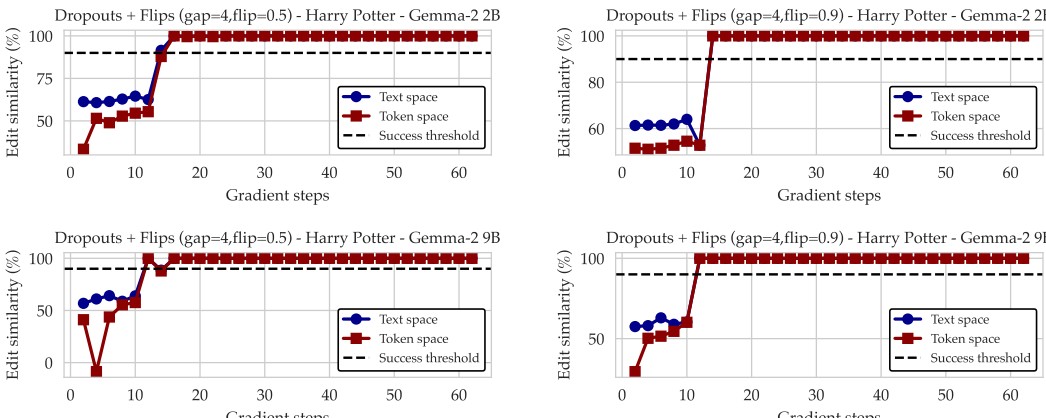

Figure 19: **Completion success on *Harry Potter Paraphrase* (Appendix B.1) for *token dropouts + casing flips* over gradient steps.** Visualizing two model sizes (Gemma-2 2B and 9B) and flip probability $p \in \{0.5, 0.9\}$. X-axis is the number of gradient steps (at batch size 32). Y-axis is the completion efficacy.

## B.7. Effect of Randomized Token Dropouts & Casing Flips on $n$-gram Overlap

Recall from Section 5 and Appendix B.4 that the **token dropouts** procedure (§5.2, Algorithm 2) and the **casing flips** procedure (§5.3, Algorithm 3) admit *randomized* versions, where every token or character is dropped or flipped with a certain probability, respectively. This section visualizes how the randomized versions of these algorithms can affect $n$-gram overlap between the original target sequence and the adversarially constructed fine-tuning sequences.

Fig. 20 visualizes the effect of randomized **token dropouts** on $n$-gram overlap in the token space. While deterministic token dropouts can guarantee no $n$-gram overlap, randomized dropouts also easily reach zero $n$-gram overlap for values starting $n \geq 6$.

Fig. 21 visualizes the effect of randomized **casing flips** on $n$-gram overlap in the token space. Due to the mechanisms of byte-pair encoding tokenization used in modern LLMs, while casing flips do not change the semantics of the original string much, it can completely alter the token space representations and result in near zero $n$-gram overlap for $n \geq 4$ (e.g., on **Karpathy**).

Fig. 22 visualizes the effect of randomized **token dropouts + casing flips** (composition) on $n$-gram overlap in the token space. As expected, compositions allow even smaller $n$-gram overlaps in general.

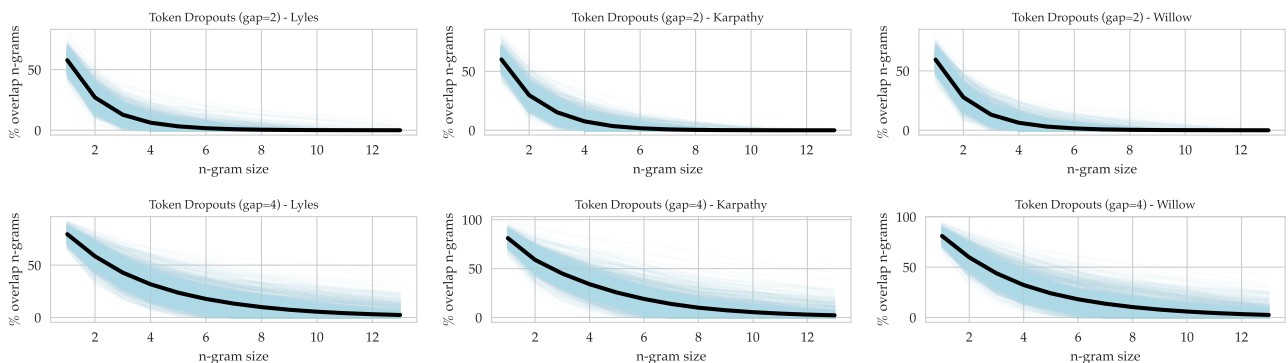

Figure 20: Amount of $n$-gram overlap between the original target sequence and the adversarially constructed fine-tuning sequences under **token dropouts** (§5.2). X-axis is the value of $n$ for $n$-gram. Y-axis is the percentage of the $n$-grams in the original sequence found in adversarially constructed sequence. Each faint **blue line** is a separate constructed fine-tuning sequence of different randomness (1000 in total), and **black line** is the average. Visualizing drop interval $d \in \{2, 4\}$ on the three main text targets (Appendix B.1).

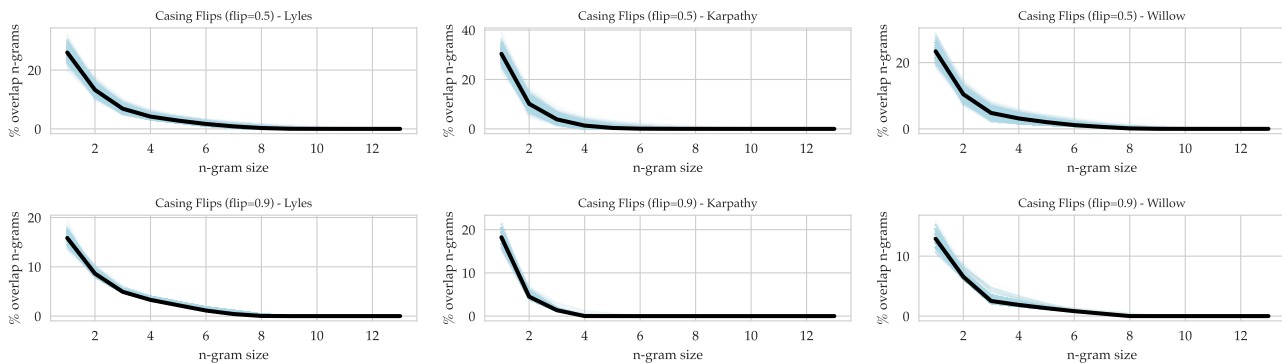

Figure 21: Amount of $n$-gram overlap between the original target sequence and the adversarially constructed fine-tuning sequences under **casing flips** (§5.3). X-axis is the value of $n$ for $n$-gram. Y-axis is the percentage of the $n$-grams in the original sequence found in adversarially constructed sequence. Each faint **blue line** is a separate constructed fine-tuning sequence of different randomness (1000 in total), and **black line** is the average. Visualizing casing flip probability $p \in \{0.5, 0.9\}$ on the three main text targets (Appendix B.1).

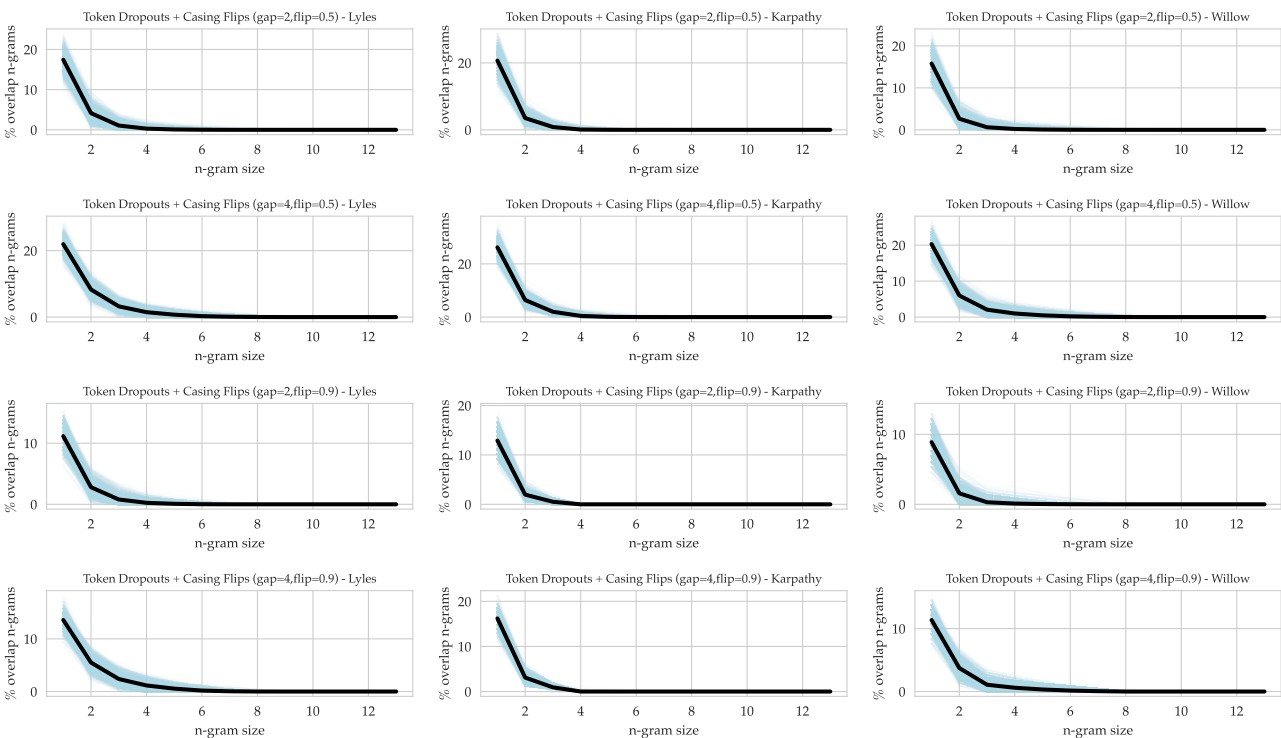

Figure 22: Amount of $n$-gram overlap between the original target sequence and the adversarially constructed fine-tuning sequences under **token dropouts + casing flips** (§5.4). X-axis is the value of $n$ for $n$-gram. Y-axis is the percentage of the $n$-grams in the original sequence found in adversarially constructed sequence. Each faint **blue line** is a separate constructed fine-tuning sequence of different randomness (1000 in total), and **black line** is the average. Visualizing drop interval $d \in \{2, 4\}$ and casing flip probability $p \in \{0.5, 0.9\}$ on the three main text targets (Appendix B.1).

