# OpenReview forum: "Language Models May Verbatim Complete Text They Were Not Explicitly Trained On"
_ICML.cc/2025/Conference — ICML 2025 spotlightposter_

### Official Review · Reviewer_7JNu · 2025-02-24

**Overall Recommendation:** 4

**Summary:**

This work shows that current LLMs are able to generalize from their training dataset in a way that completes unseen (measured by n-gram overlap) data samples during inference time, thus challenging the use of n-gram-based metrics for a wide range of fields such as memorization, contamination, poisoning, and dataset inference. The paper approaches this from two angles: (1) it shows that even when removing all sequences of verbatim copying (memorization) from an original dataset, there remain so-called "lingering sequences" that are still reproduced verbatim after full re-training (and likely an artifact of very similar neighboring sequences or natural generalizations which evade filtering) - notably this is still the case for, e.g., 5-gram filters (albeit less and with noticeable trend to generalization). Second, they show that by creating a specific dataset that shares no (n-gram) overlap with a given target sequence, a model can be "forced" to learn the target sequence verbatim (via generalization). For this, the authors test three techniques (and their combinations): chunking, dropout, and case flip. Notably, dropout (and its combinations) highlight that such sequences can be easily forced while evading filters - even after only a few fine-tuning gradient steps. The authors also find that more capable models are more susceptible (likely due to their increased capabilities in generalizing from similar sequences). Overall, the work challenges the notion of membership definitions via n-gram or exact match commonly used in various related fields.

## Update after rebuttal

The reviewer stands by their decision at the end of the reviewer rebuttal discussion and is in favor of acceptance.

**Claims And Evidence:**

All main claims are backed up by sufficient and strong empirical evidence. In particular:

- C: The threshold for n-gram-based membership in LLM training data can be gamed.
- C: n-gram definitions have limitations in the context of data membership inference.
- C: Removing n-gram members does not have to prevent LLM Verbatim Completion.
- C: One can force LLM verbatim completion by adding n-gram non-members.

have direct experiments with solid ablations.

**Essential References Not Discussed:**

None

**Experimental Designs Or Analyses:**

The reviewer has read all experimental setups including their additional descriptions in the Appendix. The definitions of algorithms have only been skimmed, but they are straightforward enough not to warrant any questions.

**Methods And Evaluation Criteria:**

Yes, across experiments and settings, both the datasets and the models are reasonable and provide a solid empirical basis for the claims made. Ablations across various parameters are reasonable and presented in an understandable format. Used models are realistic in size (given resource constraints and evaluations on the dataset itself are thorough enough to be convincing).

**Other Comments Or Suggestions:**

- L1340 otheriwse

**Other Strengths And Weaknesses:**

#### Strengths

- Extremely well-written with clear structure, definitions, and presentation. The reviewer thoroughly enjoyed reading the paper.
- The main claims of the paper are important and well substantiated with a wide range of sensible experiments showing that existing metrics can be easily circumvented.
- Ablations are reasonable and the ablations about finding closest matches and types of lingering examples are appreciated to help the overall understanding.

#### Weaknesses

- Minor in the context of this work: While the proposed techniques for forcing the memorization of a specific sequence tend to work in the "un-defended" scenario, in a realistic scenario for data poisoning (e.g., 5.6.1), it would likely be caught by a variety of filters and could be interesting follow-up directions.

##### Comments
- It might make sense to mention again in the later sections that the case flip results only hold for later normalized text.

**Questions For Authors:**

- What ways do you see to craft such sequences in a way such that they are not killed by common pre-processing techniques for model training (normalization, high-perplexity filtering)?
- Could such memorization-forcing sequences already be included in the pre-training and not only in finetuning  - is there anything that would prevent this generalization?

**Relation To Broader Scientific Literature:**

Overall, the authors of the work have made significant efforts to contextualize their work concerning the surrounding scientific literature. While the reviewer is primarily familiar with Memorization and DI literature, the key notion in many of these fields was that verbatim memorization (of a high enough entropy sequence) is very close to a gold standard of its verbatim containment in the training set. The work starts to challenge this assumption, opening up further discussions about what it means to contain data during training and finding proof of such containment - increasingly relevant topics with larger LLM adoption.

**Theoretical Claims:**

The work did not contain any theoretical claims of which one could check the soundness.

---

> ### Author Rebuttal · Authors · 2025-03-31
>
> We appreciate the reviewer’s positive assessment of our work! We’re glad the reviewer finds the paper enjoyable to read, the experiments well-designed, and the claims important for the community. We hope to address the comments below and would appreciate the reviewer’s consideration.
>
> > [Weakness] …in a realistic scenario for data poisoning (e.g., 5.6.1), it would likely be caught by a variety of filters and could be interesting follow-up directions.
>
> > [Q1] … craft such sequences in a way such that they are not killed by common pre-processing techniques for model training (normalization, high-perplexity filtering)?
>
> We appreciate the reviewer’s insightful question. In principle (recall the beginning of Sec 5), we believe that any noisy transformation of a reasonable noise distribution should work, and there should be transformations that result in “normal looking” text. For example, following **Reviewer J1BJ**’s suggestion and our response, one could consider:
> - Shuffling token ordering;
> - Multi-lingual replacement;
> - Token replacement within a restricted corpus (e.g. a fixed book, thus the perturbed sequences look like that book).
>
> We agree with the reviewer that there can be many interesting follow-up and hope to explore additional noisy transformation strategies.
>
> > [Q2] Could such memorization-forcing sequences already be included in the pre-training and not only in finetuning - is there anything that would prevent this generalization?
>
> The reviewer raises an interesting question. In **Appendix A.3**, we performed a costly edit-distance search of a few lingering sequences; judging from the visualizations, one could potentially interpret the near-duplicates and template sequences commonly in webtext as a form of “memorization-forcing sequences” that has no n-gram overlap. Given the vast size of pre-training data and the existence of lingering sequences (Sec 4), we believe that such generalization would happen naturally; however, strong data deduplication techniques that extend beyond n-gram overlap—such as semantic deduplication or MinHash/LSH deduplication that operates on very small values of n-gram—could potentially mitigate this at the pre-training level.
>
> > [Comments] It might make sense to mention again in the later sections that the case flip results only hold for later normalized text.
>
> Thank you for the suggestion! We will mention it in the updated version.

---

> > ### Comment · Reviewer_7JNu · 2025-04-02
> >
> > I thank the authors for their rebuttal and will keep my score and opinion that the paper should be accepted.

---

### Official Review · Reviewer_aZwA · 2025-03-01

**Overall Recommendation:** 4

**Summary:**

This paper illustrates numerous challenges with the existing n-gram definition of membership in the LLM privacy community. They show that LLMs can output n-grams even if they have all been removed from the training data. They then show that the n-gram definition can be gamed by constructing training datasets which do not satisfy n-gram notions of membership but still yield the target completions.

## update after rebuttal
After the rebuttal, in which I mainly asked the authors to clarify some details in the discussion, I keep my high score.

**Claims And Evidence:**

Yes, the claims are supported by the experiments.

**Essential References Not Discussed:**

N/A

**Experimental Designs Or Analyses:**

Yes, the experimental setup for all experiments seem rigorous.

**Methods And Evaluation Criteria:**

Yes, the methods used are standard in the LLM privacy literature.

**Other Comments Or Suggestions:**

It might be useful to run the same experiments on an alternative pre-training dataset, or at least include a longer discussion on the choice of it.

I think it might also be worthwhile to study how the main results on n-grams change for fine-tuning data. I would imagine that n-grams might be a better metric if the finetuned data is extremely unlikely, i.e. a hash is unlikely to be reconstructed unless it is included in the training data.

**Other Strengths And Weaknesses:**

Strengths:
1. This paper is conceptually important and of broad interest for the LLM privacy and unlearning communities.
2. The paper is very well-written and lots of intuition is provided.
3. The authors do a very solid job of rigorously exploring these "lingering" sequences in the training data and linking them with simple generalizations.

I could not find any serious weaknesses.

**Questions For Authors:**

1. Could the authors include more discussion on alternative definitions and tests for membership?
2. How would the results change in the fine-tuning regime?
3. Could you report CIs on important results, i.e. Table 1 and Figure 3?

**Relation To Broader Scientific Literature:**

Much of the existing literature on LLM privacy has relied on notions of n-gram overlap to quantify membership for the training data. However, this submission challenges the validity of this metric by providing numerous experiments that show that n-grams can be
reproduced even if all n-grams in the training data are removed.

**Theoretical Claims:**

N/A

---

> ### Author Rebuttal · Authors · 2025-03-31
>
> We thank the reviewer for the positive assessment of our work! We’re glad that the reviewer finds the work rigorous, well-written, and of broad interest to the community. We hope to address comments below, and would appreciate the reviewer’s consideration.
>
> > It might be useful to run the same experiments on an alternative pre-training dataset, or at least include a longer discussion on the choice of it.
>
> We fully agree with the reviewer. For example, it would be interesting to examine math/code-heavy datasets to observe possible new behaviors in the “lingering sequences”. We chose FineWeb since it is a state-of-the-art, open dataset, and it is sufficiently large to provide high confidence in the generalizability of our findings. Since pre-training experiments are very costly, we were unable to extend our results to other pre-training sets while maintaining a similar degree of comprehensiveness. We will add this discussion in the updated version.
>
> > It might also be worthwhile to study how the main results on n-grams change for fine-tuning data … a hash is unlikely to be reconstructed unless it is included in the training data
>
> This is an interesting direction. Indeed, one axis we did not fully explore in this work is the spectrum of “entropy” of sequences, and how it relates to completion; hashes (or random character strings) constitute the high entropy end of the spectrum. Our removal experiments (sec 4) partially addresses this: the lingering sequences that persist after applying strong filters (e.g., n-gram=5) are of “lower entropy” (following more predictable patterns) compared to sequences that persist at higher thresholds (e.g., n-gram=50).
>
> > [Q1] Could the authors include more discussion on alternative definitions and tests for membership?
>
> Thank you for the suggestion!
> - **On definition.** A key message of our work is that “membership” in LLMs extends beyond simple set membership of text in the raw dataset (Intro); it includes data neighborhoods (“soft membership”) arising from LLM generalization, data provenance, preprocessing, and other auxiliary information available throughout the training pipeline. We believe that a precise definition may require a standalone framework and supporting experiments, and consider this important future work.
> - **On tests.** Our results suggest that the “completion test” seems robust, in that it captures exact n-gram membership, near-duplicates (Sec 4), scattered information about a sequence in the training set (Sec 5), and other generalization behaviors by the LLM. We hope to explore other tests in future work.
>
> We will incorporate these discussions in the updated version.
>
> > [Q2] How would the results change in the fine-tuning regime?
>
> We assume that the reviewer refers to extending the removal experiments (sec 4, the study of “lingering sequences”) to the fine-tuning regime (sec 5). Our intuition is that our paper effectively demonstrates the remarkable ability of LLMs to generalize from “neighboring” text, where the neighborhood spans text space (e.g., Fig 8), token space (e.g., Table 21), and even semantic space. From this angle, we expect “lingering sequences” to also exist in the fine-tuning regime as long as their “neighbors” exist. We hope to explore this in future work and will add this as discussion.
>
> > [Q3] Could you report CIs on important results, i.e. Table 1 and Figure 3?
>
> We appreciate the reviewer’s suggestion! Since every point in Table 1 / Fig 3 corresponds to a pre-training run *from scratch* (33B tokens), it was computationally prohibitive to repeat all settings. That said, we report 5 repeated runs for the 1.6B parameter model in **Appendix A.9**; we find that the error bars are fairly small.

---

> > ### Comment · Reviewer_aZwA · 2025-04-01
> >
> > Thank you for your detailed rebuttal. I keep my score.

---

### Official Review · Reviewer_vJUh · 2025-03-13

**Overall Recommendation:** 2

**Summary:**

The authors present a study on the ability of LLM to generate and complete verbatim text, which they were not explicitly exposed to during training. They start by challenging the n-gram overlap membership, showing how redacting samples filtered using this criterion does not hinder LLMs' capability to generate these sequences verbatim. They then continue showing how to generate adversarial samples with no n-gram overlaps with a reference sequence that, when used for model training, allows LLMs to complete the reference sequence verbatim.

**Claims And Evidence:**

The authors rightly raise concerns about the usage of n-gram membership, as it is associated with high rates of false negatives and fails to consider broader notions of membership.
I'm less aligned with the conclusions on machine unlearning and the possibility of generating adversarial examples easily.
As reported in the manuscript, the completions and verbatim reproductions are due to the inconsistency of n-gram membership ("near duplicates, sequences with m < n-grams that are not removed or are explained by the model’s generalization capabilities"). Hence, models can reproduce completion verbatim because they interpolate using similar examples that are trivially different from target sequences.
Similarly, the generation of the adversarial samples is designed following the definition of n-gram-based membership. This membership definition is faulty and allows for generating slightly perturbed trivial samples that allow, once the model is trained on them, to interpolate "unseen" sequences.

**Essential References Not Discussed:**

I'm not aware of any essential related work not referenced.

**Experimental Designs Or Analyses:**

As hinted in previous sections on "Claims And Evidence" and "Methods And Evaluation Criteria", the experiments reported in section 5 are designed to prove a point that is somehow already addressed by showing the inconsistency of relying on n-gram membership. Moreover, the strategies proposed to generate adversarial samples (especially case-flipping) expose the model to slightly noisy versions of the sequences that are then tested for verbatim generation.

**Methods And Evaluation Criteria:**

As stated above, while the experiments on highlighting the limitations of n-gram membership are interesting, the following experiments on generative adversarial examples are somehow not surprising and do not provide additional insights.

**Other Comments Or Suggestions:**

I strongly recommend that the authors reconsider the paper structure and present the work as a perspective/position paper. After raising concerns about n-gram-based membership with the experiments they have already conducted, they could outline and present strategies to overcome such limitations. This contribution can be highly relevant to the scientific community.

**Other Strengths And Weaknesses:**

Besides what is already highlighted in previous sections, I believe the paper nicely sets the stage for a problem, i.e., the need for better membership definition when considering datasets for LLM training and evaluation, but then misses the opportunity to suggest or propose mitigation strategies or better ways to define membership.

**Questions For Authors:**

No further questions besides the comments.

**Relation To Broader Scientific Literature:**

Despite the relevance of investigating LLM behavior and concerns/risks linked to the possibility of reconstructing data, the nature of the contribution and the way it is presented have limited impact.

**Theoretical Claims:**

No theoretical claims or proofs provided.

---

> ### Author Rebuttal · Authors · 2025-03-31
>
> We appreciate the reviewer’s time and effort! We understand & address the concerns below and would appreciate the reviewer’s consideration.
>
> > [Claims & evidence] less aligned with the conclusions on unlearning
>
> We wish to clarify that:
> - Unlearning serves many goals, one is output suppression [1].
> - Our experiments (Sec 4) match the “golden baseline” of unlearning: removing target sequences and re-training **from scratch**.
> - Our conclusions on unlearning (Sec 6) directly follow **Sec 4.2**: even this golden baseline is insufficient for output suppression.
>
> The reviewer may be concerned that our data removal wasn’t thorough; in **Fig 3 / Sec 4.2** we show that this golden baseline is insufficient even under strong filters that account for many near-duplicates.
>
> [1] https://arxiv.org/abs/2412.06966
>
> > [Methods & eval] … generative adversarial examples are somehow not surprising
>
> > [Experimental design] Sec 5 experiments repeat points from n-gram membership inconsistency
>
> We clarify that our removal (Sec 4) and addition (Sec 5) experiments are **complementary** in challenging n-gram membership:
> - The insufficiency of removing n-gram members (Sec 4) illustrates **natural** “non-member” completions as part of model training (e.g., ineffective data cleaning);
> - The feasibility of adding n-gram non-members (Sec 5) illustrates **adversarial** “non-member” completions that exploit this property (e.g. hard-to-detect backdoors by the model developer, or data poison by 3rd party).
>
> We provide **concrete demonstrations and recipes** in Sec 5 in part to make limitations of n-gram membership easier to compare rigorously and reproduce across publications.
>
> > [Relation to literature] …the nature of the contribution and the way it is presented have limited impact.
>
> We emphasize that the simplicity of our findings does not take away their significance. We defer to:
> - **Reviewer 7JNu**: “The work … opens up further discussions … [for] increasingly relevant topics with larger LLM adoption.”
> - **Reviewer J1BJ**: “[this work] demonstrates that the community should strengthen the methods they use to determine data membership.”; “The paper … would be important to the community.”
> - **Reviewer aZwA**: “Much of the existing literature on LLM privacy has relied on notions of n-gram overlap”; “This paper is conceptually important and of broad interest for the LLM privacy and unlearning communities.”
>
> While the limitations of n-gram membership may be folklore knowledge, we provide the first systematic study of its failure modes which we hope is useful for the community.
>
> > [Claims & evidence] … (less aligned with) the possibility of generating adversarial examples easily …
>
> > [Experimental design] … the strategies … expose the model to slightly noisy versions of the sequences …
>
> We appreciate the reviewer’s viewpoint. We first clarify that:
> - **Adversarial sequences are easy to generate.** Simple perturbations (in both token-space and text-space) are sufficient (Sec 5).
> - **Adversarial sequences are not “trivially different”.** **Tables 19, 21 (appendix)** illustrate that “trivial” differences in **token space** can result in vast differences in **text space** due to BPE tokenization. This is important as such sequences may, e.g., evade human inspection. Note that case-flipping is but one simple, pathological choice we explore that avoids n-gram overlap in token space.
>
> In line with **Reviewer J1BJ**, we imagine many other transformations; e.g., shuffling token ordering, multi-lingual replacement, and token replacement within a restricted corpus. Our key message is that these transformations are easily extensible, allowing room for evolving adversaries and defenses.
>
> > [Other strengths and weaknesses] the paper nicely sets the stage … but then misses … mitigation strategies
>
> > [Other comments] … they could outline and present strategies to overcome such limitations.
>
> We appreciate the reviewer’s positive assessment of our experiments and framing! We hope our work is also judged by how it opens up an important yet underexplored discussion on data membership.
>
> We agree with the importance of mitigations. Our paper’s key message is that LLM “membership” extends beyond set membership of text in the raw dataset (Intro)—it also includes data neighborhoods due to generalization, provenance, preprocessing, and other auxiliary information throughout the training. A precise definition (and mitigation) may require a standalone framework and supporting experiments, which we consider as important future work.
>
> > [Other comments] Recommendation as a perspective/position paper.
>
> We thank the reviewer for the suggestion! We believe that our paper is better suited for main track since: (1) our work emphasizes systematically demonstrating the limitations of n-gram membership, rather than advocating a position; and (2) a position paper emphasizes conceptualization whereas our paper goes beyond and proposes new methodology and empirical experimentation.

---

> > ### Comment · Reviewer_vJUh · 2025-04-03
> >
> > Thank you for the detailed rebuttal.
> > I appreciate the fact that authors reiterated the main points from the paper, but my concerns on the nature of the contribution remain, so I will keep my score.

---

> > > ### Author Response · Authors · 2025-04-06
> > >
> > > Dear Reviewer vJUh,
> > >
> > > Thank you again for your engagement and feedback. We understand and appreciate your concern. We believe that we responded to your concerns point-by-point logically and faithfully (not just "reiterated the main points from the paper"), and would really appreciate if you could let us know if there’s anything else we could add for your consideration. Thank you!

---

### Official Review · Reviewer_J1BJ · 2025-03-14

**Overall Recommendation:** 5

**Summary:**

This paper demonstrates LLMs may generate verbatim versions of text that is not included in their training data *as measured by n-gram membership tests*. After demonstrating this, they examine some possible reasons why this may be and show that it is possible to adversarially inject samples into training data in such a way that they are not measured as being included by existing membership tests, but may still be generated by LLMs trained on the data.

**Claims And Evidence:**

The main claims are:

1. LLMs can generate text verbatim that n-gram membership tests fail to recognize
2. It is possible to perturb verbatim texts in such a way to obtain this behavior

My evaluation is as follows:

## 1. Verbatim generations ##
This claim is well shown. The authors demonstrate that it is possible to remove training data from models using n-gram filtering and yet still see verbatim generations of the text when prompting the model with relevant prefixes. Upon investigation they discover that this is not because the model is truly able to generate this without training data, but rather because smaller pieces not detected by the n-gram filter are still present in the training data. This is thoroughly explored across a variety of filter sizes, demonstrating the impact how filtering is done.

## 2. Training data perturbations ##
This claim is also well demonstrated. The authors devise a series of perturbations to the training data that allow target texts to still be present in the training data while remaining undetected by n-gram filtering. This includes splitting text across multiple data locations, removing chunks of text, and flipping the case of random characters. They observe models trained on this data can generate verbatim completions of the texts, despite the perturbation. They further explore the impact of how extreme the perturbations are to the LLM ability to generate completions.

**Essential References Not Discussed:**

Not that I am aware.

**Experimental Designs Or Analyses:**

The experiments appear well designed to me. Membership tests are performed across a variety of n-gram lengths, and multiple examples of completions are observed for non-members. This allows quantification of how choice of n impacts this discrepancy. When testing methods of adversarially adding data to the training set, the perturbations are intuitive and plausible (e.g. a case flipped sample would still be recognized as source text by a human, but not by a language model), demonstrating easy ways that this poisoning could occur.

**Methods And Evaluation Criteria:**

The evaluation methods are reasonable. Though the authors test on a limited set of texts when poisoning datasets, they are able to replicate results across different texts, indicating that it is likely not due to specific qualities of the text. They evaluate commonly used membership tests using different settings as well.

**Other Comments Or Suggestions:**

Addressed in other sections.

**Other Strengths And Weaknesses:**

# Strengths
The premise of the paper is well motivated and explored. The experiments detail methods to elicit the behavior as well and provide explanations for why n-gram membership tests fail. The paper is generally clear and well written and would be important to the community.

# Weaknesses
Only GPT-2 models are considered, due to training feasibility. While understandable, it would be good to see how this applies to other models, particularly those that have been fine-tuned to not generate certain text due to copyright or privacy concerns.

**Questions For Authors:**

1. How is human detectability measured for the adversarial samples?
2. Do you test shuffling the order of the base text in any experiments? It appears in the perturbations presented that the order is maintained (though dropout or distance is added), which may make the "repair" task of the model easier, but could also be a tool for detection if ordering is important.

**Relation To Broader Scientific Literature:**

This paper relates to privacy, memorization, and unlearning literature for LLMs. It contributes by examining how effective n-gram measures of membership, which are commonly used for testing whether a model has regurgitated training data, actually work. By highlighting settings where these measures fail, it demonstrates that the community should strengthen the methods they use to determine data membership.

**Theoretical Claims:**

N/A

---

> ### Author Rebuttal · Authors · 2025-03-31
>
> We appreciate the reviewer’s positive assessment of our work! We’re glad that the reviewer finds the paper well-motivated, well-written, and important to the community. We address the comments below.
>
> > [Weakness 1]: Only GPT-2 models are considered, due to training feasibility. While understandable, it would be good to see how this applies to other models, particularly those that have been fine-tuned to not generate certain text due to copyright or privacy concerns.
>
> We appreciate the reviewer’s suggestions! We’d like to note that we also explored Gemma and Qwen models for the adversarial / fine-tuning experiments (Sec 5), and we agree with the reviewer that exploring models other than GPT-2 would be useful for the pre-training experiments (Sec 4) if we had additional compute. Models that are specifically fine-tuned to avoid certain text (e.g. models that were applied post-hoc unlearning) sound interesting; we hope to explore this in future work.
>
> > [Q1]: How is human detectability measured for the adversarial samples?
>
> The reviewer raises a good question. We interpret “human detectability” as distinguishing original sequences from adversarially perturbed samples (Sec 5). Due to scope limitations, we did not conduct large-scale human studies on detectability; we primarily relied on manual inspection of generated samples (visualized in **Tables 18–21**), finding examples such as those in **Table 21** difficult to discern by us without the original context. We agree with the reviewer that it could be a valuable extension.
>
> > [Q2]: Do you test shuffling the order of the base text in any experiments?
>
> We appreciate the reviewer’s suggestion! Among all possible noisy transformations (recall the beginning of Sec 5), we explored only the three presented. Shuffling text order is indeed a valid and straightforward strategy that could easily complement our proposed transformations. More broadly, we imagine the class of transformations can be easily expanded upon; for example:
> - Chunk shuffling;
> - Multi-lingual replacement (semantic equivalents from other languages); and
> - Token replacement within a restricted corpus (e.g. a fixed book, thus the perturbed sequences look like this book).
>
> Because models get better at completions with scale (Fig 6), we hypothesize that larger models can accommodate even noisier strategies. We hope to explore in future work.

---

### Decision · Program_Chairs · 2025-05-01

**Decision:**

Accept (spotlight poster)

**Comment:**

This paper demonstrates LLMs may generate verbatim versions of text that is not included in their training data as measured by n-gram membership tests. After rebuttal, it received mixed scores of 2445. The main concern shared by the reviewer who gave the score of 2 is that the results and contributions are limited in the current form, and it would be ideal if a new metric can be proposed. On the other hand, all the other 3 reviewers have shown strong willingness to defend this work, by commenting that (1) this paper offers important conceptual and practical contributions to the field, (2) all main claims are backed up by sufficient and strong empirical evidence, (3) the paper is very well-written and lots of intuition is provided, etc. Given all these positive feedbacks, the AC would like to recommend acceptance of the paper.